# Enhance Ethanol Sensing Performance of Fe-Doped Tetragonal SnO_2_ Films on Glass Substrate with a Proposed Mathematical Model for Diffusion in Porous Media

**DOI:** 10.3390/s24144560

**Published:** 2024-07-14

**Authors:** Juan G. Sotelo, Jaime Bonilla-Ríos, José L. Gordillo

**Affiliations:** School of Engineering and Sciences, Tecnologico de Monterrey, Eugenio Garza Sada 2501, Monterrey 64849, Mexico; jgsotelof@tec.mx (J.G.S.); jlgordillo@tec.mx (J.L.G.)

**Keywords:** sol–gel, Fe, dip-coating, SnO_2_, gas sensor, model, synthesis, ethanol, dopants, tin oxide

## Abstract

This research enhances ethanol sensing with Fe-doped tetragonal SnO_2_ films on glass, improving gas sensor reliability and sensitivity. The primary objective was to improve the sensitivity and operational efficiency of SnO_2_ sensors through Fe doping. The SnO_2_ sensors were synthesized using a flexible and adaptable method that allows for precise doping control, with energy-dispersive X-ray spectroscopy (EDX) confirming homogeneous Fe distribution within the SnO_2_ matrix. A morphological analysis showed a surface structure ideal for gas sensing. The results demonstrated significant improvement in ethanol response (1 to 20 ppm) and lower temperatures compared to undoped SnO_2_ sensors. The Fe-doped sensors exhibited higher sensitivity, enabling the detection of low ethanol concentrations and showing rapid response and recovery times. These findings suggest that Fe doping enhances the interaction between ethanol molecules and the sensor surface, improving performance. A mathematical model based on diffusion in porous media was employed to further analyze and optimize sensor performance. The model considers the diffusion of ethanol molecules through the porous SnO_2_ matrix, considering factors such as surface morphology and doping concentration. Additionally, the choice of electrode material plays a crucial role in extending the sensor’s lifespan, highlighting the importance of material selection in sensor design.

## 1. Introduction

The detection and quantification of ethanol gas are essential across many sectors, including industry, environmental monitoring, and healthcare [1]. Accurate and reliable ethanol gas sensing technology ensures safety, enhances quality control, and supports health diagnostics [2,3,4]. In recent years, significant strides have been made in developing ethanol sensors, aiming to achieve higher sensitivity, selectivity, and reliability in detecting ethanol in complex gas mixtures [5,6].

Among the diverse materials explored for ethanol sensing applications, thin tetragonal tin dioxide (SnO_2_) films have emerged as promising candidates [7]. SnO_2_ films offer several advantageous properties, including high surface area [8], chemical stability [9], and sensitivity to reducing gases [10]. These characteristics make SnO_2_ films attractive for ethanol sensing, as they enable the efficient adsorption and detection of ethanol molecules, thus facilitating accurate and rapid ethanol detection. However, despite their inherent properties, pristine thin SnO_2_ films often exhibit limitations in terms of sensitivity and selectivity, particularly at low ethanol concentrations and in the presence of interfering gases [11].

To address these challenges, researchers have turned to various strategies such as surface functionalization, morphological control, and, notably, doping [12,13,14]. Doping involves introducing foreign elements, like transition-metal ions, into the SnO_2_ lattice to enhance its properties and sensor performance [15,16]. Among the dopants investigated for enhancing ethanol sensing, iron (Fe) has garnered significant attention due to its unique characteristics and potential synergistic effects when incorporated into the SnO_2_ matrix [17,18,19]. Fe dopants offer distinct advantages for ethanol sensing applications. Fe nanoparticles boast high surface area, catalytic activity, and magnetic properties, which can enhance sensitivity and selectivity by facilitating interactions with ethanol molecules and promoting charge transfer processes [20,21]. Integrating iron (Fe^+3^) dopants into the SnO_2_ lattice is expected to synergistically enhance ethanol sensing performance, offering improved sensitivity, selectivity, and response time compared to undoped SnO_2_ films [22,23].

This study builds on previous research by Fe^+3^ doping SnO_2_ films specifically for ethanol detection. While other transition-metal ions such as Cu [24], Mn [25], and Co [26] have also been used to dope SnO_2_, iron (Fe) doping provides a unique combination of catalytic and electronic properties that are particularly beneficial for ethanol sensing. This allows for a more detailed investigation into how Fe influences the sensor’s dynamic response characteristics compared to other dopants [27,28].

Beyond industrial and environmental applications, Fe-doped SnO_2_ films’ enhanced ethanol sensing capabilities have promising implications for disease diagnosis [29]. Breath analysis has gained attention as a non-invasive and rapid diagnostic tool for various diseases [30], including diabetes [31], asthma [32], and lung cancer [33]. Elevated ethanol levels in breath samples have been associated with certain metabolic disorders and respiratory conditions, making ethanol sensors valuable for disease screening and monitoring [34,35,36]. By leveraging the enhanced sensitivity and selectivity of Fe-doped SnO_2_ films, these sensors have the potential to revolutionize disease diagnosis, offering early detection and personalized treatment strategies.

In this study, we present a comprehensive investigation into the enhanced ethanol sensing performance of Fe-doped tetragonal SnO_2_ films synthesized via the sol–gel method. We evaluate the ethanol sensing performance of the doped films through electrical measurements, highlighting the influence of dopant type and concentration on sensitivity and response time. This study contributes to a deeper understanding of dopant-mediated enhancements in SnO_2_-based ethanol sensors and offers valuable insights into designing and optimizing sensing materials.

We aim to address the limitations of current ethanol sensing technologies by enhancing the sensitivity and selectivity of SnO_2_ films through iron (Fe) doping. Despite extensive research on SnO_2_-based sensors, significant challenges remain, particularly in achieving high performance under complex environmental conditions. By integrating Fe, known for its catalytic and magnetic properties, into SnO_2_ films, we hypothesize that the interaction dynamics with ethanol molecules can be significantly improved, thereby enhancing sensor performance. This research investigates the effects of Fe doping on the fundamental properties of SnO_2_ films and explores the potential of Fe-doped SnO_2_ films in practical applications, ranging from industrial safety to disease diagnosis through breath analysis.

The findings presented herein hold significant implications for developing efficient and reliable ethanol-sensing devices, with potential applications in automotive emissions control, industrial process monitoring, and breath analysis. These insights are expected to contribute to the design and optimization of advanced sensing materials, potentially leading to the development of more efficient and reliable ethanol sensors.

## 2. Materials and Methods

### 2.1. Synthesis of the Samples

Two SnO_2_ sensors were manufactured; one following the method outlined in a previous work [7], and the other doped with Fe^+3^, which will be named from now on in this document as the SnO_2_-Fe sensor. Initially, a solution consisting of 50 mL of 0.05 mol SnCl_4_·5H_2_O and 50 mL of 0.5 M citric acid solution is prepared and stirred for 15 min to ensure thorough mixing. A predetermined amount of (Fe(NO_3_)_3_·9H_2_O) is added to the mixture. The quantity is carefully controlled to achieve the desired concentration and effect, which is 4:1 for this work. Subsequently, the solution is stirred for an additional 15 min to achieve homogeneity. Then, 50 mL of a 2% (*w*/*v*) aqueous Polyethylene glycol (PEG) solution (Mw = 1450 Da), previously stirred for 60 min, is added to the tin chloride, citric acid, and iron (III) nitrate nonahydrate solution. This mixture is stirred for another 15 min. The gradual dropwise addition of concentrated ammonium hydroxide (NH_4_OH) follows until the desired pH of 11 is attained, resulting in a distinctive milky brownish color. Using ammonium hydroxide in the synthesis is beneficial because it volatilizes upon heating, leaving only tin and iron compounds behind. This ensures a pure final product by eliminating potential contaminants.

Notably, including these salts induced a discernible transformation in the color profile of the resultant aqueous solution as illustrated in Figure 1, transitioning from a milky whitish hue to a distinctive brownish shade. Beyond color alteration, this process via the sol–gel method offers several advantages. One notable benefit is its adaptability, allowing for seamless integration with diverse techniques and strategies, thereby enabling the possible addition of various transition metals to synthesize novel chemical components.

The synthesis of the stannic chloride pentahydrate (SnCl_4_·5H_2_O) and citric acid, with the addition of iron (III) nitrate nonahydrate and subsequent pH adjustments using ammonium hydroxide, involves several chemical reactions [37,38,39]:(1)$$Sn{Cl}_{4}+4C_{6}H_{8}O_{7}⟹Sn(C_{6}H_{5}{O_{7})}_{4}+4HCl+4H_{2}O.$$

The reaction between tin chloride and citric acid forms a tin citrate complex, producing hydrochloric acid [40]. During the addition of PEG, no specific chemical reaction takes place. Instead, PEG serves as a stabilizing agent in the synthesis process [41]. PEG serves multiple functions in this synthesis process, contributing to particle size and iron (Fe) distribution improvements [42]. It prevents the agglomeration or coalescence of nanoparticles during formation, thus stabilizing particle size and ensuring a more uniform distribution [43].
(2)$$Sn(C_{6}H_{8}O_{7}{)}_{4}+19O_{2}⟹SnO_{2}+24CO_{2}+{16H}_{2}O,$$

Then, the decomposition of the tin citrate complex at elevated temperatures leads to the formation of tin (IV) oxide nanoparticles, carbon dioxide, and water [44].
(3)$${NH}_{4}OH+HCL⟹NH_{4}Cl+H_{2}O,$$

Finally, the ammonium hydroxide reacts with the remaining hydrochloric acid from the previous step to form ammonium chloride and water [45]. The reaction between iron (III) nitrate and the other components in the solution could potentially lead to the formation of iron-containing species, contributing to synthesizing the sensor material. The chemical reaction involving iron (III) nitrate in the synthesis process is:(4)$$Fe(NO_{3}{)}_{3}+3C_{6}H_{8}O_{7}⟹Fe(C_{6}H_{5}O_{7}{)}_{3}+3HNO_{3},$$
here, iron (III) nitrate reacts with citric acid to form an iron citrate complex and nitrate acid [46]. The iron citrate complex could potentially participate in the subsequent steps of the synthesis process, contributing to forming tin (IV) oxide nanoparticles doped with iron.

### 2.2. Dip-Coating and Tempered

Two separate sensors were manufactured using an identical coating methodology. In the sensor fabrication process, the cylindrical hollow glass substrate underwent thorough cleaning with acetone and isopropanol before being meticulously coated inside the sol–gel solution. The dip-coating procedure demanded careful attention to technical variables such as immersion and withdrawal speed to ensure a uniform coating [47,48]. Both sensors were coated with an immersion speed of 10 mm/s and a withdrawal speed of 5 mm/s.

Following coating, the substrates were dried at 110 °C for 1 h in ambient air and then heat-treated at 400 °C for 10 min in a furnace. This iterative process was repeated six times for both sensors to achieve the desired coating thickness. Upon completing the sixth iteration, each sample underwent further heat treatment at 600 °C for 1 h. The final high-temperature treatment was employed to transform the material into the specific tetragonal rutile form, known for its effectiveness in detecting ethanol gas. The tetragonal SnO_2_ phase (rutile-type, mineral cassiterite) is the only stable polymorph of SnO_2_, whereas other phases are metastable. Due to its stability, the tetragonal SnO_2_ phase is primarily studied as a gas-sensing material [49]. This study focuses on the stable tetragonal phase to leverage its well-documented properties and reliability for gas sensing applications. This phase is widely recognized in the literature for its superior ethanol sensing properties.

### 2.3. Characterization

X-ray diffraction was performed on a Miniflex 600 Powder X-ray Diffractometer (Rigaku, Monterrey, Mexico). The measurements were performed with a source voltage of 30 kV and 15 mA using a Cu cathode (Kα) as a source at a wavelength of 1.5418 Å. The diffraction patterns were from 20 to 80° with a step of 0.05° and a scanning rate of 2°/min. No sample preparation was required. Nitrogen adsorption–desorption isotherms were obtained in a Quantachrome Autosorb-iQ analyzer. Before measuring, the samples were degassed at 60 °C for 1 h, 120 °C for 2 h, then at 200 °C for 4 h with ramps of 10 °C/min; at the end of the degassing process, a test was carried out to confirm the degassing, which concluded satisfactory in all cases. Finally, scanning electron microscopy (SEM) and energy-dispersive X-ray spectroscopy (EDX) images were taken using an EVO MA25 field-emission scanning electron microscope (Zeiss, Monterrey, Mexico), operating at an acceleration voltage of 30 kV.

### 2.4. Sensor Assembly

Following the characterization phase, attention turns towards the mechanical and electrical assembly of the sensors. Considerations include electrode placement for efficient gas detection, incorporating temperature control mechanisms for precise operation and using suitable materials to withstand operational stresses.

Several adaptations have been made to ensure the better functionality of the sensors for mechanical assembly. Two sensors, the SnO_2_ with copper electrodes and the SnO_2_-Fe sensor with silver-plated gold electrodes, were mechanically and electrically assembled. Both sensors were mounted on glass-reinforced epoxy laminate material. This setup was designed to compare the lifespan and stability of both sensors against each other. Both sensors were subjected to a one-month period of reheating at 120 °C separately to observe their signal behavior and allow for stabilization. The sensing element is configured within a voltage divider circuit, and an external potential source is used to adjust and regulate the operating temperature of both sensors as illustrated in Figure 2.

### 2.5. Gas Test Chamber

A transparent acrylic box was created to characterize the experiments. It featured a flow inlet and several holes for gas exit. The experimental set was proposed to contain two air pumps (Figure 3a,b) that are responsible for injecting the same amount of air pressure into both: a mixing acrylic box (Figure 3c) and a glass Erlenmeyer flask with a bolt neck (Figure 3d).

The air introduced into the flask is directed under pressure to the water–alcohol mixture using a syringe. This allows the mixture to bubble and generate gaseous ethanol. These bubbles leave the neck of the flask towards the mixing box (where it is mixed with the air from the first air pump) and, thus, can be mixed homogeneously. In the end, the turbulence-filled mixture is injected into the acrylic test chamber (Figure 3e), where it can finally be sensed by all the sensors that are statically positioned (Figure 3f) and send the obtained signal to a personal computer where it can be analyzed (Figure 3g). This controlled environment allows for the precise regulation of gas concentration, humidity, and temperature, mimicking real-world conditions. The chamber enables the validation of sensor responses to specific gases, optimizing sensitivity to gaseous ethanol.

Maintaining a relative humidity of over 35% is widely recognized as optimal for sampling [50]. A relative humidity sensor ensured the real-time monitoring and maintenance of humidity levels within the 40% to 45% range. This validation suggests the potential of the sensor presented in this study to detect trace ethanol concentrations in human breath [51]. Detecting such low levels of ethanol is particularly significant for monitoring health conditions like diabetes, where ethanol can be an indicator of abnormal metabolic processes, such as those occurring in diabetic ketoacidosis [52,53,54].

## 3. Results and Discussion

### 3.1. Structure Analysis

The X-ray diffraction (XRD) pattern was utilized to ascertain the phase and purity of both the SnO_2_ and the SnO_2_-Fe sensors. The observed peaks in the XRD patterns closely matched the standard values obtained from the JCPDS file no. 41-1455 [55]. These values correspond to a lattice parameter of a = 4.738 Å and c = 3.187 Å for tin dioxide. The XRD Smart Lab Studio II software identified all diffraction peaks, confirming the rutile-type tetragonal crystal structure (cassiterite) in the SnO_2_ and SnO_2_-Fe sensors. Figure 4 illustrates their respective XRD patterns.

Introducing iron (Fe) salts into the synthesis of SnO_2_ films might potentially affect the crystallography observed in X-ray diffraction (XRD) analysis. The presence of Fe dopants may lead to changes in the crystal lattice structure of SnO_2_, altering the position and intensity of the diffraction peaks. Additionally, incorporating Fe dopants could influence the width of the peaks in the XRD pattern. This explains that the peaks become broader than the original SnO_2_ sensor, reflecting changes in crystallite size, strain, or defects within the material.

The XRD software successfully identified all diffraction peaks. The Debye–Scherrer equation was used to estimate average crystallite sizes [56]:(5)$$D=\frac{0.9\;\lambda }{\beta \cos\theta },$$
where l is the X-ray wavelength, b is the full width at half maximum, and q is the corresponding angle. The average size of SnO_2_ is 12.9 nm while that of SnO_2_-Fe is 24.50 nm. The broadening of XRD peaks is generally associated with smaller crystallite sizes, as described by the Debye–Scherrer equation. However, peak broadening can also result from other factors such as strain, defects, and instrumental effects [57]. In our study, we observed broader peaks for SnO_2_-Fe compared to SnO_2_, yet the crystallite size calculated for SnO_2_-Fe was larger (24.5 nm) than that for SnO_2_ (12.9 nm). This apparent contradiction suggests that the broadening of peaks in SnO_2_-Fe is not solely due to smaller crystallite sizes but may also be influenced by additional factors such as strain or defects introduced by Fe^+3^ doping. It has been reported that the α-phase of Fe_2_O_3_ is obtained by annealing the samples at 400 °C [58]. It is suggested that the peaks of Fe_2_O_3_ could be in the same region as the SnO_2_ peaks. These factors can contribute to the broadening of the diffraction peaks without necessarily indicating a smaller crystallite size. There are two possible options: they crystallize separately, or the iron is within the crystalline structure of the rutile crystal of the SnO_2_.

### 3.2. Superficial Area

Obtaining nitrogen adsorption–desorption isotherms is crucial for characterizing materials’ surface area, porosity, and pore size distribution. The SnO_2_ sensor exhibits notable surface and pore characteristics, with a specific surface area of 28.73 m^2^/g. This surface area suggests enhanced reactivity and capacity, making it suitable for applications such as adsorption and catalysis. The pore size distribution analysis reveals a mode pore size of 17.28 nm, indicating a predominant pore size range within the material. This information is crucial for understanding the material’s performance in filtration and gas separation processes. Furthermore, the material possesses a pore volume of 0.11 cc/g, indicating its porosity and capacity for the adsorption and diffusion of gases or liquids.

Figure 5a exhibits H_1_-type hysteresis, typically associated with uniform materials with spherical particles. Despite this, SnO_2_ shows a complex pore structure. This complexity arises not from the particle shape itself but from how these spherical particles are arranged and packed together. The uniformity in particle shape can lead to varied interparticle spacing, especially when particle aggregation occurs in different manners or under varying synthesis conditions. This aggregation can create diverse pore sizes and shapes, leading to the observed irregular adsorption behavior [58]. This type of hysteresis loop characterizes materials with cylindrical pore geometry and a certain level of pore size uniformity [59].

H_1_-type is commonly observed in materials composed of compacts of nearly spherical particles arranged in a relatively uniform manner [60]. The observed variation in the form of the H_1_-type loop may be attributed to findings from a prior study where SEM imaging revealed a morphology resembling a forest of randomly oriented flakes. These random pathways are ubiquitous, indicating a substantial number of reactive areas attributable to the high porosity of the material. Figure 5a (inset) shows the adsorption process as an irregular process ascending and descending every certain period until reaching a peak and then a gradual but constant decay period. The irregular behavior in the plot may be Influenced by the surface morphology and structure of the material. If the material consists of randomly oriented flakes on the surface, it can result in a complex pore structure with irregular adsorption behavior. The irregularities in the plot could be attributed to variations in the accessibility and availability of adsorption sites within the pores, which may differ depending on the orientation and arrangement of the flakes.

The SnO_2_-Fe sample exhibited a surface area of 17.70 m^2^/g, indicating a substantial area of exposed surfaces per gram of the sample. The analysis of the pore size distribution revealed a mode pore size of 8.78 nm, suggesting that pores around this size are prevalent within the sample. The presence of pores within this size range could enhance the sample’s performance in applications like gas storage or separation processes. The pore volume was also measured to be 0.05 cc/g, indicating a considerable volume of pores available for adsorption or storage. These results provide valuable insights into the structural properties of the SnO_2_-Fe sample, which can be essential for understanding its behavior in various applications, such as catalysis, gas adsorption, or materials synthesis.

Figure 5b (inset) depicts the adsorption process. The plot shows the derivative of cumulative pore volume concerning pore diameter. It starts with a steady ascent, indicating the presence of pores of various sizes contributing to the total volume. It peaks at a pore diameter of approximately 5 × 10^−3^ nm, emphasizing the predominant pore size or range. Then, it steadily decreases without significant noise, suggesting a decline in the contribution of larger pores to the total volume. This indicates that the material’s pore sizes are distributed in a single peak.

Figure 5b shows H3-type hysteresis, indicative of slit-like pores commonly found in plate-like particles. Despite the presence of a wide range of pore sizes as suggested by the H3 hysteresis, the pore size distribution depicted in Figure 5b predominantly shows a single peak. This observation can be explained by the dominant pore size formed due to the primary mode of particle aggregation or Fe doping levels, which influences the overall pore structure. Although there is a predominant peak, smaller and larger pores also exist but are less frequent, contributing to the overall complexity of the pore structure. The IV(a)-type isotherm typically exhibits a gradual increase in adsorption at low relative pressures, followed by a sharp rise at higher pressures, indicating the presence of mesopores or macropores within the material [61]. This suggests adsorption occurring via monolayer formation on the external surface and within smaller pores initially, transitioning to multilayer adsorption within larger pores as pressure increases [62]. The hysteresis loop type H3 indicates a reversible adsorption–desorption process with a wide range of pore sizes or a complex pore structure [63]. In type H3, the desorption branch is above the adsorption branch at all relative pressures, suggesting desorption occurs at higher pressures than adsorption, possibly due to capillary condensation within larger pores.

In an ethanol sensor, the IV(a)-type isotherm and hysteresis loop type H_3_ would exhibit behavior indicative of efficient ethanol molecule detection. Initially, at low ethanol concentrations, the sensor’s response gradually increases, suggesting adsorption occurring through monolayer formation on the sensor’s surface and within smaller pores. As ethanol concentration rises, the response sharply increases, indicating multilayer adsorption within larger pores. This signifies the presence of mesopores or macropores within the sensor material, facilitating effective ethanol molecule detection [64]. The hysteresis loop type H_3_ reflects reversible adsorption–desorption processes, where even as ethanol concentration decreases, the sensor maintains a signal output, implying the retention of some ethanol molecules within the pores or active sites. The desorption branch above the adsorption branch indicates desorption at higher pressures, likely due to capillary condensation in the pores. These characteristics are vital for the sensor’s precise, long-term detection of ethanol concentrations.

### 3.3. Morphological Analysis

In our previous work [7], the SnO_2_ films demonstrated a forest-like morphology with randomly oriented flakes, enhancing porosity and reaction areas due to the formation process involving multiple dip-coating cycles and sol–gel synthesis. Similarly, Figure 6a of the current study shows an SEM image of the SnO_2_-Fe sensor, where the fractal and porous characteristics are apparent. The fractures within the metal oxide area are produced through the dip-coating and sintering cycle. The particles appear to be irregularly shaped with sharp edges and angular features, suggesting that the sample may have undergone mechanical fragmentation through manipulation or crystallized in an angular form. The surface of the particles appears rough and porous, which could be beneficial for applications requiring high surface area, such as sensing. The particles seem to be aggregated, forming clusters, likely due to the processing method or the natural tendency of SnO_2_-Fe particles to agglomerate. The particles’ high surface area and porous nature can enhance the sensitivity and response time of gas sensors made from this material. The rough surface and high surface area are also advantageous for catalytic reactions, providing more active sites for reactions. The variation in contrast observed in the SEM image of the SnO_2_-Fe sample (Figure 6b) is influenced by several factors related to the sample’s morphology and imaging conditions. The observed contrast differences are primarily due to variations in particle size and the presence of different phases, such as iron oxide. Larger particles or aggregates tend to backscatter electrons more strongly, appearing brighter in the image [65]. This is influenced by the higher average atomic number of the materials forming these aggregates compared to the SnO_2_ matrix. Additionally, factors like particle shape and surface roughness also play significant roles in how electrons are backscattered, further contributing to the contrast variations seen in SEM images.

In the energy-dispersive X-ray spectroscopy (EDX) image of the SnO_2_-Fe sample, the variations in brightness correspond to differences in the concentration and distribution of elements within the material [66]. In Figure 6c, brighter regions in the elemental maps indicate areas with higher concentrations of specific elements. For example, in the tin (Sn) map (Figure 6d), the brighter areas signify regions with a higher density of tin, likely representing SnO_2_-rich areas. In the oxygen (O) map (Figure 6e), brighter regions correlate with higher oxygen content, which may align with regions of concentrated SnO_2_ or other oxides. Similarly, in the iron (Fe) map (Figure 6f), the brighter regions indicate higher iron concentrations, suggesting iron-rich phases or pure iron areas.

The SEM analysis reveals variations in brightness, suggesting compositional heterogeneity at a microscale within the sample. These variations highlight regions where iron might appear more concentrated or prevalent due to local differences in particle size, shape, or aggregation. Conversely, the EDX analysis provides a broader perspective, indicating that iron is homogeneously distributed within the SnO_2_ matrix across the entire sample. This uniform distribution, suggesting excellent diffusion behavior, ensures that there are no significant concentrations or deficiencies of iron throughout the material. Such a homogeneous dispersion of iron is beneficial for the performance of SnO_2_-Fe composites as gas sensors. The even distribution of iron atoms enhances the material’s sensitivity and response time by providing uniformly dispersed active sites for gas interaction, thus ensuring consistent and reliable sensor performance.

### 3.4. Gas Sensing Experiment

The experiment was conducted as follows: the equipment was run for three and a half minutes with only room air to obtain a stable signal. Then, alcohol vapors and room air were mixed in a small chamber and introduced into the testing chamber, allowing the mixture to flow for three minutes. Subsequently, the flow of ethanol vapors was stopped, and only air was allowed to flow for five minutes to achieve de-saturation. This cycle was repeated up to four times for each of the three runs to observe signal repeatability, resulting in a total of twelve curves for each combination of temperature and concentration.

This data acquisition process was applied to both sensors under the same conditions to allow for comparison. After four reversible cycles, signal data were acquired. To compare the operational work in these new batches of sensors, resistance in air and gas was measured. Table 1 shows the gas response of the SnO_2_ sensor at different temperatures and concentrations. The sensitivity of the sensor was systematically investigated by analyzing its gas response values (S), defined as the ratio of resistance in dry air (Ra) to resistance in the presence of gas (Rg) in reducing gases such as ethanol [67]. It should be noted that, for convenience, the signal was flipped in our measurements, which means that Rg appears higher than Ra. The data indicate a general trend of increasing sensitivity with both temperature and gas concentration, though some deviations from this trend were observed.

The general trend observed across the data indicates that the SnO_2_ sensor’s sensitivity improves with increasing temperature and gas concentration. This suggests that the sensor becomes more efficient and responsive at higher temperatures and higher gas concentrations. The highest sensitivity values recorded at 120 °C indicate that the sensor’s optimal temperature for maximum sensitivity lies in this range. The consistent increase in sensitivity with gas concentration at all tested temperatures indicates that the SnO_2_ sensor can effectively differentiate between different levels of gas presence. This is crucial for applications requiring precise gas detection and quantification.

At first glance in Table 2, the observation that the resistance in the air is more stable in the SnO_2_-Fe sensor compared to the undoped SnO_2_ sensor implies that the presence of Fe dopant may influence surface phenomena. This stability suggests that the Fe dopants alter the surface chemistry of the SnO_2_ material, leading to enhanced stability in the presence of air. One possible mechanism through which this stabilization occurs is the promotion of oxygen adsorption on the surface of the SnO_2_-Fe sensor by the Fe dopants, forming a more stable oxide layer. This stabilized oxide layer could contribute to the consistent resistance observed in the air.

The data also indicate that the SnO_2_-Fe sensor exhibits different sensitivity characteristics than the undoped SnO_2_ sensor, with notable implications for practical applications. In Figure 7a, the SnO_2_-Fe sensor shows higher sensitivity at lower temperatures (80 °C) and reduced sensitivity at higher temperatures (100 and 120 °C), suggesting that Fe doping alters the temperature response of the sensor. This contrasts with the undoped SnO_2_ sensor, demonstrating increased sensitivity at elevated temperatures. The reduced sensitivity of the SnO_2_-Fe sensor at higher temperatures may be due to enhanced ethanol desorption, leading to a diminished interaction with the sensor surface. Consequently, the SnO_2_-Fe sensor may be more suitable for applications requiring lower operating temperatures, while the undoped SnO_2_ sensor is better suited for high-temperature environments where maximum sensitivity is critical; these findings underscore the importance of selecting the appropriate sensor material and operating conditions based on the specific requirements of the gas detection applications.

The graph comparing the sensor response of SnO_2_ and SnO_2_-Fe sensors to ethanol gas at 120 °C across various concentrations in Figure 7b reveals significant differences in their performance. The SnO_2_ sensor exhibits a constant, low response of around 10 across all ethanol concentrations, indicating minimal sensitivity to ethanol concentration changes at 120 °C. In contrast, the SnO_2_-Fe sensor shows markedly higher sensitivity, with the response increasing with ethanol concentration up to 20 ppm, peaking around 50, and then slightly decreasing at 40 ppm.

Figure 7c illustrates the gas response of SnO_2_ and SnO_2_-Fe sensors to ethanol gas at 120 °C over time, highlighting the enhanced ethanol sensing performance of the Fe-doped SnO_2_ sensor. The SnO_2_ sensor exhibits a relatively low and stable response of 7.2 to 40 ppm of ethanol, indicating minimal sensitivity to ethanol at this concentration and temperature. In contrast, the SnO_2_-Fe sensor shows a significantly higher response to 1 ppm ethanol, reaching peaks at around $4.6{\times 10}^{3}$ kΩ, demonstrating substantial sensitivity even at much lower ethanol concentrations compared to the SnO_2_ sensor. The response pattern of the SnO_2_-Fe sensor, characterized by periodic peaks and troughs, indicates a strong and consistent response to repeated ethanol exposure.

The Fe-doped SnO_2_ sensor’s high sensitivity at low ethanol concentrations and robust and repeatable performance underscores its superior ethanol detection capabilities. The sharp rise and fall of the sensor’s response peaks suggest a quick reaction to ethanol exposure and rapid recovery when the gas is removed, which is crucial for real-time monitoring applications. The SnO_2_-Fe sensor demonstrates exceptional ethanol sensing performance across various concentrations (1, 5, 10, and 20 ppm), significantly outperforming the undoped SnO_2_ sensor. In Figure 7d, at 1 ppm, the SnO_2_-Fe sensor exhibits a substantial response, indicating high sensitivity even at very low concentrations. This sensitivity remains robust at 5 and 10 ppm, showcasing its reliability and consistent performance. At 20 ppm, the sensor achieves peak response, underscoring its efficiency in detecting moderate ethanol levels.

Response and recovery times (t_res_ and t_rec_) are key parameters in gas sensor assessment. Response time indicates how quickly a device detects a target compound, influencing its responsiveness [68]. Conversely, recovery time reflects the system’s readiness for repeated measurements, vital for assessing sensing-system throughput, especially in medical and agrifood screening analyses [69]. Analyzing the SnO_2_ sensor response and recovery time to ethanol detection in Table 3 reveals important insights into its functionality. As the operating temperature increases, the sensor’s response time improves, thereby detecting ethanol more quickly. However, higher temperatures also lead to longer recovery times, indicating the sensor takes more time to reset and be ready for the next detection. The ethanol concentration affects recovery time more noticeably than response time, but its impact is less significant than temperature changes. The sensor is most effective for rapid detection at higher temperatures, though there is a trade-off with slower recovery. These findings highlight the importance of selecting an optimal operating temperature to balance quick detection and efficient recovery, ensuring reliable performance for practical applications.

Then, the data regarding the SnO_2_-Fe sensor’s response and recovery times to ethanol detection in Table 4 unveil its operational characteristics. Unlike the SnO_2_ sensor, the SnO_2_-Fe sensor exhibits consistent response times across different temperatures and ethanol concentrations, indicating a robust and less temperature-sensitive performance. The response time remains relatively stable at 80 °C, suggesting the sensor can detect ethanol regardless of environmental conditions.

Similarly, the recovery time of the SnO_2_-Fe sensor shows minimal variation, implying its resilience to changes in ethanol concentration and operating temperature. This stability in recovery time underscores the sensor’s capacity to swiftly return to its baseline state after exposure to ethanol, ensuring a dependable and predictable response in successive detection cycles.

These features imply that the SnO_2_-Fe sensor is highly effective for detecting low ethanol concentrations, making it suitable for breath analyzers, industrial safety, and environmental monitoring applications [70]. The rapid response and recovery times further enhance its suitability for dynamic sensing environments where quick detection and reset cycles are needed. Overall, the significant improvement in ethanol sensing performance due to Fe doping highlights the potential of SnO_2_-Fe sensors for advancements in gas sensing technologies, offering precise and reliable detection compared to undoped SnO_2_ sensors.

To enhance the response and recovery times of SnO_2_ and SnO_2_-Fe sensors beyond temperature adjustments, several alternative strategies can be considered. Surface functionalization with catalytic nanoparticles such as platinum (Pt) or palladium (Pd) can accelerate the adsorption and desorption processes of molecules on the sensor surface [71]. Developing sensors with nanostructured morphologies, such as nanowires, nanotubes, or nanosheets, can increase the surface area and enhance gas diffusion, leading to faster interactions between the gas molecules and the sensor surface [72]. Additionally, doping with other elements such as cobalt (Co), nickel (Ni), or copper (Cu) can create more active sites and improve the electronic properties of the sensors, facilitating quicker response and recovery [73]. Light activation using ultraviolet (UV) or visible light can enhance the desorption of gas molecules, thereby reducing recovery times and increasing sensitivity [74]. Implementing a pulsed voltage rather than a constant voltage can periodically refresh the sensor surface, promoting quicker adsorption and desorption cycles [75]. Finally, optimizing the sensor design by reducing the thickness of the sensing layer or improving the porosity can enhance gas diffusion and reaction rates, leading to faster sensor dynamics. These strategies offer promising avenues for further improving the performance of SnO_2_ and SnO_2_-Fe sensors.

Table 5 summarizes the sensitivities, concentrations, operating temperatures, response times, and recovery times for different sensor materials, including SnO_2_ and SnO_2_-Fe from this work. In comparison to other materials documented in the literature, the SnO_2_ sensor from this study demonstrates a moderate sensitivity of 7.2 at 40 ppm and 120 °C, with a response time of 47 s and a recovery time of 272 s for ethanol detection. While these values are respectable, they do not outperform certain advanced materials like SnO_2_-CdS, which exhibits an exceptionally high sensitivity of 130 at 100 ppm and 200 °C, with a shorter recovery time of 3 s. Additionally, rGO/SnO_2_ shows a significantly higher sensitivity of 48.4 at 50 ppm and 120 °C. However, the recovery and response times for rGO/SnO_2_ are not provided, making a complete comparison challenging. Despite these comparisons, the SnO_2_ sensor in this study provides a balanced performance with good sensitivity and reasonable response and recovery times, making it a viable candidate for ethanol detection.

The SnO_2_-Fe sensor developed in this study stands out by demonstrating substantial improvements in sensitivity and response characteristics compared to the undoped SnO_2_ sensor and other materials in the literature. The SnO_2_-Fe sensor achieves a sensitivity of 21.7 at just 1 ppm and 80 °C, indicating a remarkable enhancement due to Fe doping. This sensor also exhibits a response time of 73 s and a recovery time of 214 s, which are competitive with other advanced materials. For instance, In_2_O₃@SnO_2_ shows a slightly higher sensitivity of 22.6 at 100 ppm and 320 °C but with a faster response time of 1 s and a recovery time of 132 s. The γ-Fe_2_O₃/SnO_2_ sensor, while operating at 160 °C, has a lower sensitivity of 5.0 at 100 ppm but benefits from shorter response and recovery times of 25 and 11 s, respectively. Overall, the SnO_2_-Fe sensor developed in this study offers a significant advantage in terms of sensitivity at lower concentrations and lower operational temperatures, making it highly suitable for practical ethanol sensing applications.

### 3.5. Mathematical Model

In typical metal oxide sensors (MOX), the detection mechanism involves several key steps: the diffusion of the target gas molecules to the sensor surface, the adsorption of these molecules onto active sites, a chemical reaction occurring at the surface, the desorption of the reaction products, and subsequent diffusion away from the sensor [82]. This behavior has been accurately described in mathematical models applied to these MOX sensors and put into four stages to imitate the olfaction system [83]. However, the unique structure of the sensors developed in this study, which incorporate a porous medium and layers of flakes, necessitates a different behavior in these steps. The intricate morphology of these film sensors, characterized by high porosity and layered flakes, alters the pathways and dynamics of gas diffusion, adsorption, and desorption. Consequently, a more complex mathematical model is required to accurately capture the sensor’s response, considering the enhanced surface area and the multilayered structure that influences the interaction between gas molecules and the sensor material.

The schematic in Figure 8 depicts the complex processes of diffusion and chemical reactions within a detailed porous medium, characterized by a fractal-like structure. As gaseous ethanol interacts with the film surface, it can decompose into various subproducts through interactions with active regions occurring simultaneously within the sensor. Once ethanol molecules are adsorbed onto the active sites, the sensor achieves a steady state. Previous research has suggested that in such complex porous media, anomalous diffusion with adsorption in a fractal dimension is likely to occur [84]. These phenomena have been extensively explored, with various scenarios proposed, including the introduction of a logarithmic diffusion equation tailored for porous media that incorporates a time-dependent source term [85]. Consequently, to accurately model the behavior of the SnO_2_ and SnO_2_-Fe film sensors, it is essential to conduct a thorough analysis of these processes, considering the sensor as a complex porous medium.

The equation governing the change in concentration ρ is
(6)$$\frac{\partial p}{\partial t}=D\frac{\partial^{2}}{\partial x^{2}}\ln\rho -\propto \left(t\right)\rho ,$$
Thus, the solution of Equation (6) is
(7)$$\rho \left(x,t\right)=\hat{\rho }\left(x,y\right)e^{-\int_{0}^{t}\propto (t^{\prime })dt^{\prime }},$$
where
(8)$$\hat{\rho }\left(x,t\right)=\frac{1}{Z\left(\tau \left(t\right)\right)[1+\beta (\tau \left(t\right))x^{2}]},$$
where Z and β are related to the diffusivity coefficient that changes with time and position and τ is related to time. On the other hand, the source term could be considered for this research as the changing concentration of ethanol on the surface sensor:(9)$$\int_{0}^{t}\propto \left(t^{\prime }\right)dt^{\prime }=C,$$
this change in concentration is due to the oxidation of ethanol, assumed to be a first-order reaction where $C_{0}$ is the initial concentration of ethanol:(10)$$C=C_{0}e^{-kt},$$
therefore,
(11)$$\int_{0}^{t}\propto \left(t^{\prime }\right)dt^{\prime }=C_{0}e^{-kt},$$
this changing concentration in Equation (11) can be related to the consumption of electrons at the surface used to decompose ethanol into its subproducts. The substitution of Equation (11) into Equation (7) results in an equation that mathematically describes the physicochemical process within the sensor:(12)$$\rho \left(x,t\right)=\hat{\rho }\left(x,t\right)e^{-C_{0}e^{-kt}}.$$

The latter equation provides a mathematical description of how the concentration of ethanol changes, or in other words, how electrons are produced and how the electrical current evolves in a complex porous medium. However, the values of the parameters indicated by $\hat{\rho }$ (related to the change in diffusivity with time and position) are unknown and obtaining them is beyond the scope of this work. It is also important to mention that the change in ethanol concentration is not the only chemical reaction occurring on the sensor; further research is needed to derive an appropriate expression for the second exponential term. Therefore, it can be said that the exponential change in concentration is influenced by the time needed for the ethanol to fill the testing chamber. Ideally, the testing chamber can be saturated within nine seconds after the ethanol flows in. Consequently, it can be concluded that the exponential concentration term is related to the sensor filling time. Then, the logarithmic diffusion in Equation (12) may explain the complex behavior exhibited by the sensor, which is very similar to the Gompertz equation below:(13)$$f\left(x\right)=Ae^{-Be^{-Ct}},$$
where A is an asymptote value, B represents a positive displacement along the x-axis, and C sets the growth rate. The sigmoid function describes a growth pattern characterized by initial exponential growth, a middle period showing a gradual change, and a final stage of a designated mature phase. This parameterization suggests a future asymptote on the right side with a more gradual progression compared to its approaches towards the lower value asymptote from its beginning [86]. Despite the complexity, this adapted mathematical model, derived from biological growth models [87,88], can elucidate the chemical and physical phenomena underlying the behaviors observed in the graphs generated by these SnO_2_ and SnO_2_-Fe sensors, as discussed earlier.

The proposed mathematical model consists of four stages, whichh provide a framework suitable for sensors manufactured using the proposed methodology Equations (14)–(17). These stages are the start ($r_{1}$), rise ($r_{2}$), sample ($r_{3}$), and decay ($r_{4}$) periods, each described by specific equations that outline their contributions to the overall model:(14)$$r_{1}\left(t\right)=R_{0},$$
(15)$$r_{2}\left(t\right)=R_{0}+Ae^{-Be^{-Ct}},$$
(16)$$r_{3}\left(t\right)=R_{0}^{\prime },$$
(17)$$r_{4}\left(t\right)=R_{0}^{\prime }-Ae^{-Be^{-Ct}}.$$

The fitting was conducted with the objective of comparing the SnO_2_ sensor’s performance against the SnO_2_-Fe sensor and other commercial MOX sensors. For the model, a plateau was established instead of an ascending curve. This plateau, set at 90% of the maximum value, has been reported by other types of sensors as its top signal limit. As predicted, an exponential relationship explains the adsorption, chemical reaction, and desorption phenomena on the sensor’s surface. It is assumed that commercial MOX sensors were originally designed to produce a similar ascending curve; however, this feature was subsequently removed through the implementation of electronic filters within the components of the common commercial MOX sensor.

After applying the proposed model to the SnO_2_ sensor data, the model fits well at all measured temperatures and adequately represents the behavior of adsorption and desorption. Figure 9a shows the sensor’s raw data at different temperatures, demonstrating good performance in the fitting across the temperature variations. In Figure 9b, a commercial MOX sensor is also fitted using the same model function with the constants’ value presented above, indicating that this model can be used as a reference for future comparisons with other sensors in the literature.

Table 6 shows that for adsorption, parameters A and B increase with concentration, while parameter C remains constant. The slower desorption process is represented by the lower values of B and C compared to the adsorption values, which might be due to the higher polarity of the substances produced during the decomposition of ethanol (acetaldehyde, carbon dioxide, and water).

The mathematical model was also tested using different temperatures (Table 7), provided that it can be correctly applied to the SnO_2_ sensor. If the model accurately describes the behavior of the SnO_2_ sensor across different temperatures and concentrations, it suggests two key points.

Firstly, the sensor’s response to ethanol follows a consistent pattern that can be effectively characterized by the Gompertz function, regardless of variations in the temperature of ethanol concentration. This consistency in sensor behavior across different environmental conditions indicates robustness and reliability in the sensor’s response. Secondly, the successful application of the model across various temperatures and concentrations underscores the versatility and adaptability of the sensor in diverse operating environments. It suggests that the sensor’s response dynamics are not overly influenced by external factors such as temperature variations or changes in ethanol concentrations, allowing for reliable and accurate measurements under different conditions.

The root mean square error (RMSE), lower than 7, indicates that the model fits well, not only to the SnO_2_ sensor’s experimental data at different temperatures but also to that of the commercial MOX sensor. In the context of the Gompertz model, RMSE represents the average magnitude of error between the actual response data and the response values predicted by the model.

The application of the model to the SnO_2_-Fe sensor at various temperatures (Figure 10) and concentrations (Figure 10b–d) provides valuable insights into its sensing capabilities. By analyzing the sensor’s response using this mathematical framework, the dynamic behavior of the sensor concerning different environmental conditions can be elucidated. The Gompertz model, with its sigmoidal shape, aptly captures the non-linear relationship between the sensor output and the concentration of target gases across a range of temperatures. Through this analysis, critical points such as the detection threshold, saturation levels, and the rate of response can be discerned, offering a comprehensive understanding of the sensor’s performance under diverse operating conditions.

The application of the Gompertz model to the SnO_2_-Fe sensor is demonstrated by its strong fit to experimental data, highlighting its effectiveness in capturing the complex relationship between sensor response and gas concentrations (Table 8) at different temperatures (Table 9). Furthermore, with root mean square error (RMSE) values around 30, the model’s predictive accuracy is noteworthy, indicating minimal deviation between predicted and observed values. This precision is essential for ensuring reliable and consistent sensor performance, reinforcing confidence in its capability to accurately detect and quantify target gases.

The consistency and robust fit of the model to the SnO_2_, SnO_2_-Fe, and commercial MOX sensors suggest a fundamental similarity in their sensing mechanism and response dynamics. This consistency highlights the model’s applicability in proving a comprehensive and reliable characterization of sensor behavior.

Ultimately, the successful application of the model to different sensors demonstrates its effectiveness in capturing the essential dynamics of gas sensing. The model not only offers a detailed understanding of the adsorption and desorption processes but also provides a reliable framework for comparing the performance of various sensors. This approach fosters further advancements in sensor technology, ensuring that future sensors can be accurately modeled, and their performance optimized for diverse applications.

In our study, we applied an exponential model to fit the dynamic response curves of Fe-doped SnO_2_ thin-film sensors. This approach allowed us to investigate the influence of Fe doping on sensor performance, specifically analyzing parameters A, B, and C. Parameter A represents the initial response magnitude, reflecting the baseline resistance change upon ethanol exposure. Parameter B is the rate constant for the exponential growth phase, indicating the response speed of the sensor. Parameter C is the rate constant for the exponential decay phase, representing the sensor’s recovery speed after ethanol exposure is removed. These parameters are crucial for understanding the sensor’s behavior, as Fe doping alters the electronic properties and surface chemistry of SnO_2_, enhancing adsorption sites and influencing interaction dynamics with ethanol molecules.

Our findings indicate that Fe-doped SnO_2_ sensors exhibit increased sensitivity and faster response times due to the catalytic and magnetic properties of Fe. This enhances the adsorption and desorption rates of ethanol, impacting parameters A, B, and C. Specifically, Fe doping can lead to higher values of A due to increased adsorption sites and higher values of B and C due to improved response and recovery speeds, respectively. These effects are particularly significant for polar gas molecules like ethanol, as the altered electronic properties of Fe-doped SnO_2_ enhance the interaction with polar molecules.

In contrast to other studies [89], which focused on thick-film sensors with metal sandwich or co-planar electrodes and presented only a single plot of fractional conductance change, our work extends the application of the exponential model to thin-film sensors. Additionally, we provide a comprehensive analysis of the gas resistance signals and a detailed interpretation of the model parameters. This distinction highlights the novelty of our approach and its relevance in advancing the understanding of Fe-doped SnO_2_ sensors for ethanol detection.

### 3.6. Sensor Stability

Finally, the stability of sensors over time is a critical aspect of their performance and reliability in various applications, as illustrated in Figure 11. Specifically, for sensors made of different materials such as SnO_2_ and SnO_2_-Fe, the manufacturing process and choice of electrode materials significantly impact their long-term stability. The electrode material influences the sensor’s electrical properties, chemical reactivity, and resistance to environmental factors over time.

Sensors manufactured with Cu electrodes, like the SnO_2_ sensor, may exhibit different stability characteristics than those manufactured with silver-plated gold electrodes, such as the SnO_2_-Fe sensor. The superior conductivity and corrosion resistance of gold can enhance the stability of sensors with silver-plated gold electrodes over time. Conversely, sensors with Cu electrodes may be more susceptible to oxidation or chemical reactions that could degrade their performance if not properly protected or treated.

Stability considerations are vital for assessing sensor lifespan and reliability, especially in harsh or demanding environments. Long-term stability minimizes drift, calibration drift, or performance degradation, reducing the need for frequent re-calibration or replacement. Ensuring stability is crucial for maintaining consistent and reliable measurements over extended periods. In applications such as environmental monitoring, industrial process control, or medical diagnostics, sensor stability ensures accurate and repeatable results over time, improving data quality and confidence in the measurement outcomes [90].

## 4. Conclusions

The synthesis method used in this study is highly adaptable for manufacturing sensors with various dopants, demonstrated here with the incorporation of iron (Fe). The flexibility of this synthesis process allows for precise doping, tailoring the sensor properties to specific applications. EDX confirmed the homogeneous distribution of Fe throughout the SnO_2_ matrix, ensuring consistent performance across the entire sensor. Additionally, the morphological analysis indicated that the Fe-doped SnO_2_ has a surface structure conducive to efficient gas sensing, enhancing its ability to interact with ethanol molecules.

The enhanced gas response of the SnO_2_-Fe sensor compared to the undoped SnO_2_ sensor is a significant finding. The Fe-doped sensor exhibits a marked improvement in sensitivity, particularly at low ethanol concentrations and lower temperatures. This heightened sensitivity is critical for applications such as breath analyzers, where detecting volatile organic compounds at low concentrations is essential for early diagnosis and monitoring. The ability of the SnO_2_-Fe sensor to perform effectively at low temperatures also broadens its potential application scope, making it a versatile tool in various environmental and industrial settings.

Overall, the SnO_2_-Fe sensor’s consistent and reliable performance across diverse conditions positions it as a trustworthy option for ethanol detection applications. While it may not achieve the rapid response times observed with the SnO_2_ sensor at higher temperatures, its predictability and stability make it a preferred choice where consistent operation is crucial.

The successful application of the Gompertz model to the SnO_2_, SnO_2_-Fe, and the other MOX sensors provides valuable insights into the adsorption and desorption processes within the sensor’s porous structure. By incorporating the concept of diffusion in a complex porous medium with tortuosity, the model effectively captures the dynamics of gas sensing. The ability of the model to fit experimental data across various temperatures and concentrations demonstrates its robustness and versatility. This mathematical framework not only enhances our understanding of the sensors’ behavior but also offers a reliable method for predicting sensor performance under different conditions. These insights contribute to the development of more accurate and reliable models, optimizing gas sensor technology for a wide range of applications.

Furthermore, the choice of electrode material is crucial for the longevity and stability of gas sensors. In this study, the use of Fe-doped SnO_2_ improved sensitivity and suggested potential benefits in prolonging the sensor’s lifespan. The robust performance and enhanced durability of the SnO_2_-Fe sensor underscore the importance of selecting appropriate materials for electrode fabrication. This advancement highlights the potential for developing long-lasting, high-performance gas sensors, contributing significantly to the fields of environmental monitoring, industrial safety, and medical diagnostics.

## Figures and Tables

**Figure 1 sensors-24-04560-f001:**
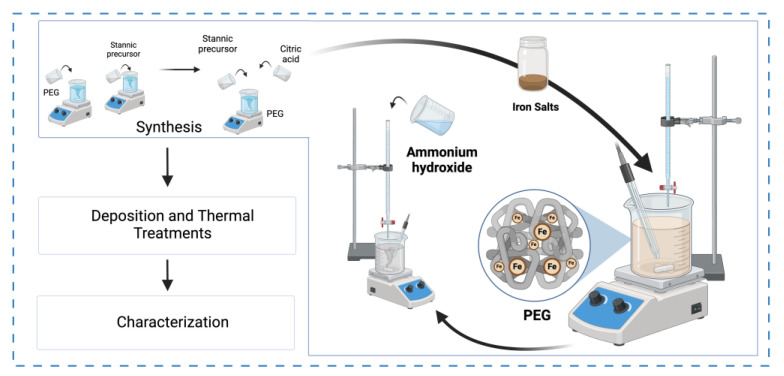
Schematic representation of Fe-doped SnO_2_ synthesis pathway.

**Figure 2 sensors-24-04560-f002:**
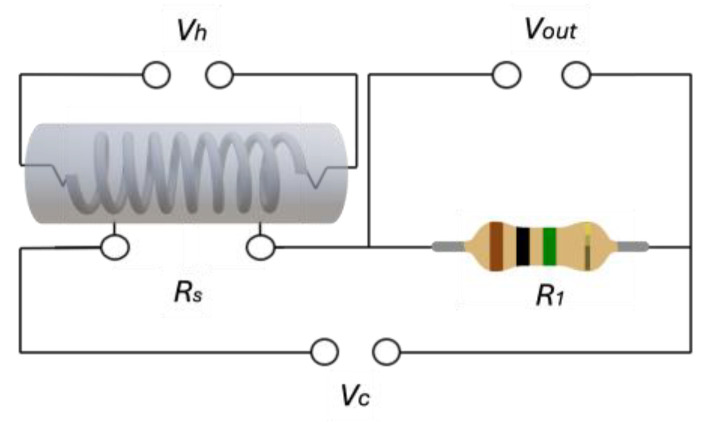
This is a general illustration of the electrical circuit for both sensor devices, where V_c_ is the circuit input voltage, V_h_ is the external heating voltage, and V_out_ is the output voltage passing through the reference resistance (R_1_). This voltage divider was designed to measure the change in the sensor resistance (R_S_).

**Figure 3 sensors-24-04560-f003:**
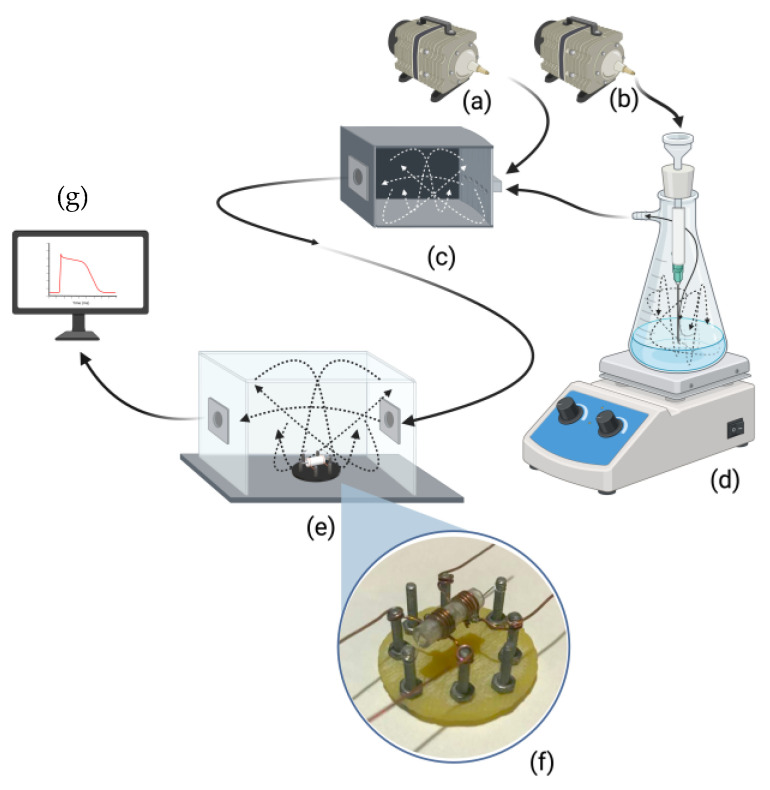
General diagram of the gas test chamber: (**a**,**b**) air pumps; (**c**) mixing box; (**d**) a glass Erlenmeyer flask with a bolt neck; (**e**) an acrylic test chamber; and (**f**) a gas sensor.

**Figure 4 sensors-24-04560-f004:**
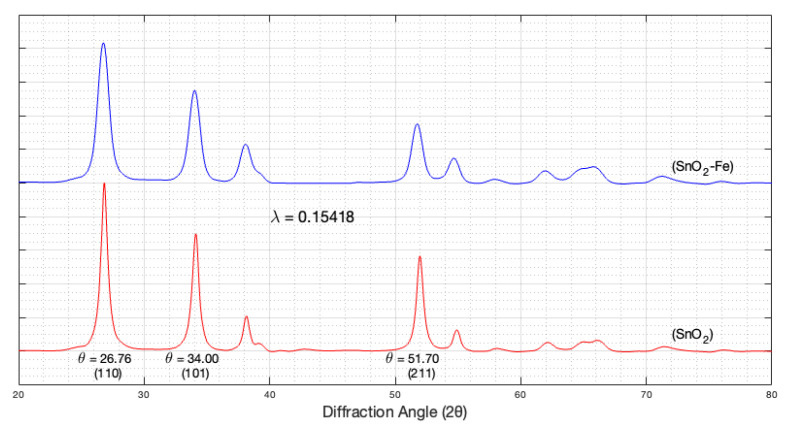
XRD patterns of the SnO_2_ and SnO_2_-Fe sensors.

**Figure 5 sensors-24-04560-f005:**
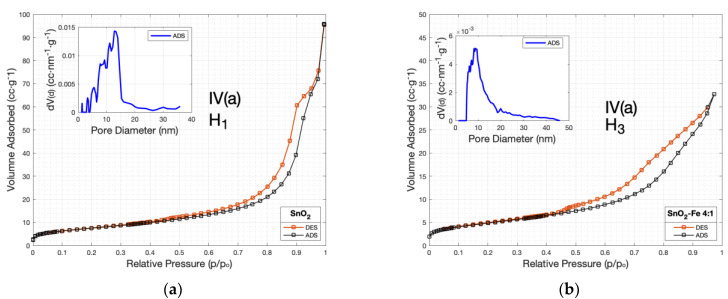
BET analysis of the adsorption−desorption behavior of (**a**) SnO_2_ and (**b**) SnO_2_-Fe.

**Figure 6 sensors-24-04560-f006:**
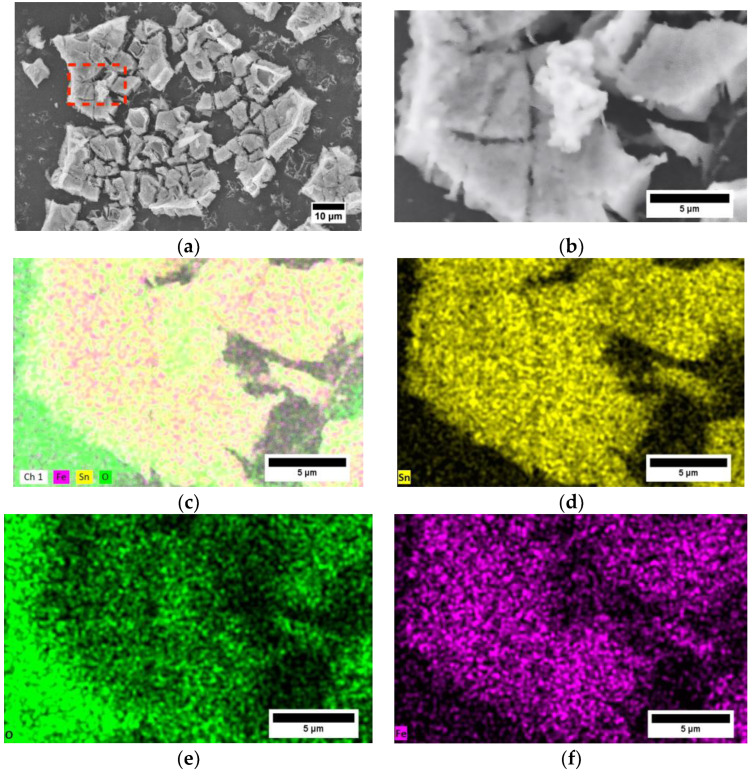
(**a**) SEM analysis of the SnO_2_ coating, and EDX analysis of the elements within the material: (**b**) SnO_2_-Fe section to analyze (red box amplification), (**c**) mapping of the elements within the material, (**d**) mapping of tin (Sn) in the material, (**e**) mapping of oxygen (O) in the material, and (**f**) mapping of iron (Fe) distributed in the material.

**Figure 7 sensors-24-04560-f007:**
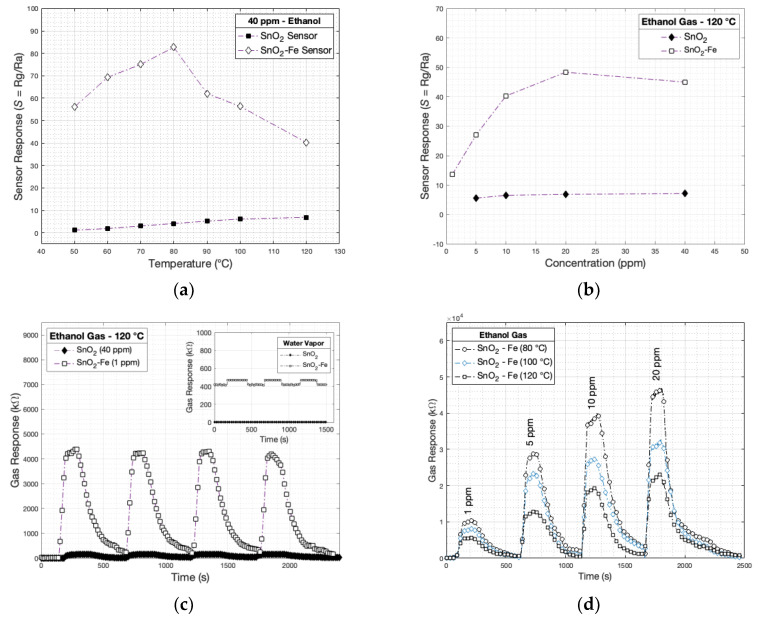
(**a**) The SnO_2_ and SnO_2_-Fe sensor response at gaseous ethanol at different temperatures and (**b**) at different concentrations; (**c**) the SnO_2_ and SnO_2_-Fe dynamic cycle response at 100 °C and (**d**) their dynamic response at different concentrations.

**Figure 8 sensors-24-04560-f008:**
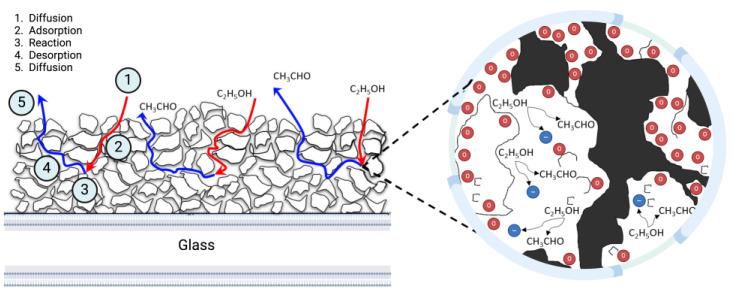
The stages involved in a porous medium within the SnO_2_ and SnO_2_-Fe sensors.

**Figure 9 sensors-24-04560-f009:**
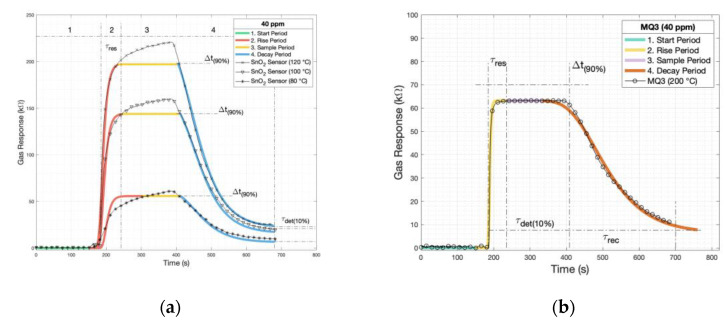
(**a**) The mathematical model applied to the SnO_2_ sensor and (**b**) the model applied to a commercial MOX sensor.

**Figure 10 sensors-24-04560-f010:**
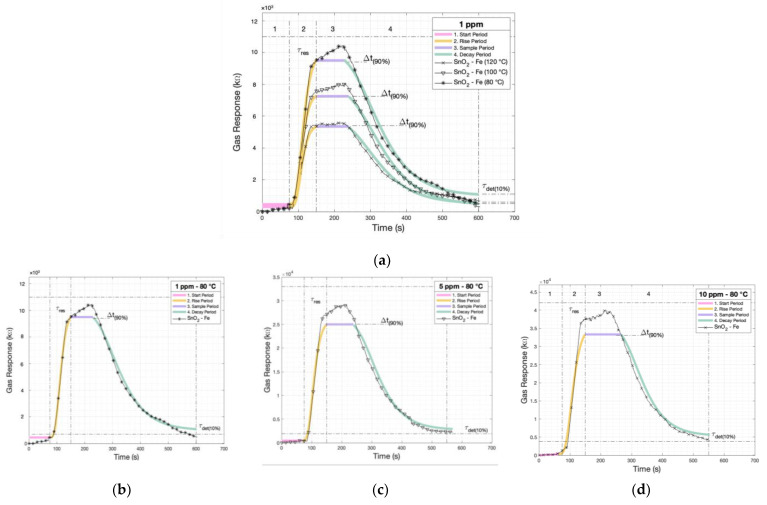
(**a**) The model applied to the SnO_2_-Fe sensor for different temperatures at 1ppm and (**b**–**d**) the model applied to the SnO_2_-Fe sensor at different concentrations.

**Figure 11 sensors-24-04560-f011:**
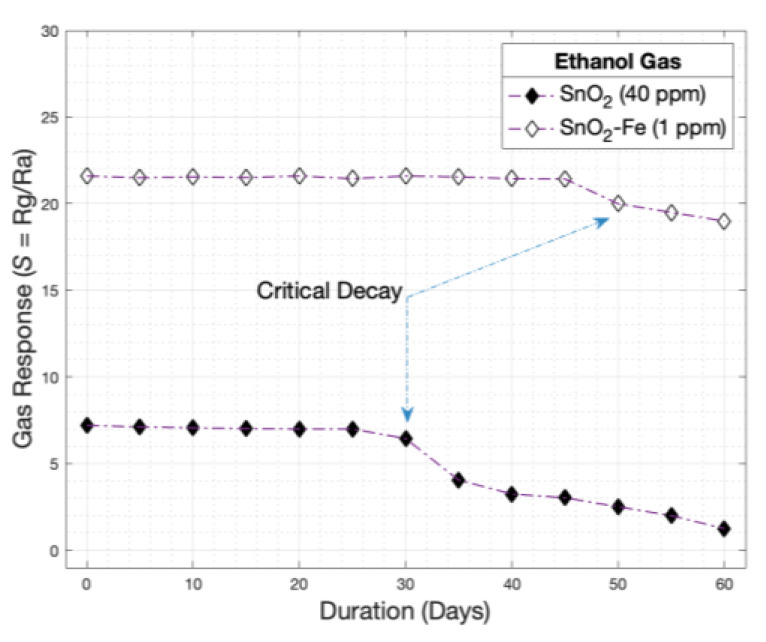
Stability of the SnO_2_ and SnO_2_-Fe sensors at different temperatures for 60 consecutive days.

**Table 1 sensors-24-04560-t001:** Minimum and maximum SnO_2_ sensor resistance in dry air (Ra) and gas (Rg).

Temperature	10 ppm			20 ppm			40 ppm		
	Ra	Rg	S ^1^	Ra	Rg	S	Ra	Rg	S
80 °C	9.5	36.8	3.8	11.9	50.7	4.3	14.1	62.3	4.4
100 °C	22.4	109.1	4.8	25.2	155.9	6.2	25.0	160.6	6.4
120 °C	29.3	194.0	6.6	11.9	82.7	6.9	30.5	220.1	7.2

^1^ S: The gas responses (S = Ra/Rg for reducing gas).

**Table 2 sensors-24-04560-t002:** Minimum and maximum SnO_2_-Fe sensor resistance in clean air (Ra) and ethanol (Rg).

Temperature	Ra	1 ppm		5 ppm		10 ppm		20 ppm		40 ppm	
		Rg	S ^1^	Rg	S	Rg	S	Rg	S	Rg	S
80 °C	$4.8{\;\times \;10}^{2}$	${1.0\times 10}^{4}$	21.7	$2.9{\;\times \;10}^{4}$	60.2	${4.0\times 10}^{4}$	82.7	$4.6{\;\times \;10}^{4}$	97.2	${3.8\times 10}^{4}$	79.0
100 °C	$4.6{\;\times \;10}^{2}$	${8.1\times 10}^{3}$	17.7	${2.3\times 10}^{4}$	48.5	${2.7\times 10}^{4}$	56.4	${3.3\times 10}^{4}$	68.1	$2.8{\;\times \;10}^{4}$	57.0
120 °C	${4.1\times 10}^{2}$	$4.6{\;\times \;10}^{3}$	11.2	${1.3\times 10}^{4}$	27.1	$1.9{\;\times \;10}^{4}$	40.3	$2.3{\;\times \;10}^{4}$	48.3	$2.2{\;\times \;10}^{4}$	45.0

^1^ S: The gas responses (S = Ra/Rg for reducing gas).

**Table 3 sensors-24-04560-t003:** SnO_2_ sensor’s response and recovery times.

Concentration	80 °C		100 °C		120 °C	
	t_res_ ^1^ (s)	t_rec_ ^2^ (s)	t_res_ (s)	t_rec_ (s)	t_res_ (s)	t_rec_ (s)
5 ppm	79	73	73	295	53	220
10 ppm	74	71	71	292	53	223
20 ppm	84	69	69	281	47	271

^1^ t_res_: Response time and ^2^ t_rec_: Recovery time.

**Table 4 sensors-24-04560-t004:** SnO_2_-Fe sensor’s response and recovery times.

Concentration	80 °C		100 °C		120 °C	
	t_res_ ^1^ (s)	t_rec_ ^2^ (s)	t_res_ (s)	t_rec_ (s)	t_res_ (s)	t_rec_ (s)
1 ppm	73	214	77	217	78	218
5 ppm	80	192	78	213	80	220
10 ppm	85	208	81	217	84	222

^1^ t_res_: Response time and ^2^ t_rec_: Recovery time.

**Table 5 sensors-24-04560-t005:** Comparison of ethanol sensing performance of various sensor materials for detecting gaseous ethanol reported in the literature.

Sensor Material	S ^1^	C ^2^	T ^3^	Tres ^4^	Trec ^5^	References
ZnO	3.2	675	100	1270	2470	[6]
SnO_2_	7.2	40	120	47	272	[7]
γ-Fe_2_O_3_/SnO_2_	5.0	100	160	25	11	[23]
SnO_2_/Mxene	5.0	10	230	14	26	[76]
⍺-Fe_2_O_3_	-	12.5	300	10	60	[77]
SnO_2_-CdS	130	100	200	21	3	[78]
rGO/SnO_2_	48.4	50	120	-	-	[79]
ZrO_2_/Co_3_O_4_	1.1	20	200	56	363	[80]
In_2_O_3_@SnO_2_	22.6	100	320	1	132	[81]
SnO_2_-Fe	21.7	1	80	73	214	This Work

^1^ S: Sensitivity, ^2^ C: Concentration (ppm), ^3^ T: Temperature (°C), ^4^ tres: Response time (s), and ^5^ trec: Recovery time (s).

**Table 6 sensors-24-04560-t006:** The mathematical model parameters for the rise and decay stages for the SnO_2_ sensor at different concentrations.

120 °C	C ^1^	Phenomena	RMSE ^2^	A _(kΩ)_	B	C _(1/t)_
SnO_2_	10	Rise	6.48	174	$5{.2\times 10}^{8}$	$1.1{\times 10}^{-1}$
		Decay	6.52	174	$6.9{\times 10}^{3}$	$2.0{\times 10}^{-2}$
	20	Rise	6.48	186	$5.6{\times 10}^{8}$	$1.1{\times 10}^{-1}$
		Decay	6.18	186	$7.3{\times 10}^{3}$	$2.0{\times 10}^{-2}$
	40	Rise	6.15	198	$5.7{\times 10}^{8}$	$1.1{\times 10}^{-1}$
		Decay	6.05	198	$7.7{\times 10}^{3}$	$2.0{\times 10}^{-2}$

^1^ C: Concentration (ppm) and ^2^ RMSE: Root Mean Square Error (kΩ).

**Table 7 sensors-24-04560-t007:** The parameters for the rise and decay stages for the SnO_2_ sensor and a commercial MOX gas sensor using the Gompertz mathematical model.

40 ppm	T ^1^	Phenomena	RMSE ^2^	A _(kΩ)_	B	C _(1/t)_
SnO_2_	80	Rise	6.95	56	$1.8{\times 10}^{9}$	$1.1{\times 10}^{-1}$
		Decay	6.15	56	$9.2{\times 10}^{3}$	$2.0{\times 10}^{-2}$
	100	Rise	6.99	144	$7.9{\times 10}^{8}$	$1.1{\times 10}^{-1}$
		Decay	6.10	144	$8.9{\times 10}^{3}$	$2.0{\times 10}^{-2}$
	120	Rise	6.15	198	$5.7{\times 10}^{8}$	$1.1{\times 10}^{-1}$
		Decay	6.05	198	$7.7{\times 10}^{3}$	$2.0{\times 10}^{-1}$
MOX Sensor	200	Rise	1.43	63	$5.7{\times 10}^{28}$	$3.5{\times 10}^{-1}$
		Decay	1.61	63	$5{.5\times 10}^{2}$	$1.3{\times 10}^{-2}$

^1^ T: Temperature (°C) and ^2^ RMSE: Root Mean Square Error (kΩ).

**Table 8 sensors-24-04560-t008:** The mathematical model parameters for the rise and decay stages for the SnO_2_-Fe sensor at different concentrations.

80 °C	C ^1^	Phenomena	RMSE ^2^	A _(kΩ)_	B	C _(1/t)_
SnO_2_-Fe	1	Rise	23.3	$9.7{\times 10}^{3}$	${1.2\times 10}^{3}$	${6.6\times 10}^{-2}$
		Decay	27.3	$9.7{\times 10}^{3}$	${9.9\times 10}^{1}$	${1.8\times 10}^{-2}$
	5	Rise	29.4	$2.6{\times 10}^{4}$	${8.1\times 10}^{3}$	${6.6\times 10}^{-2}$
		Decay	16.9	$2.6{\times 10}^{4}$	${2.2\times 10}^{2}$	${1.8\times 10}^{-2}$
	10	Rise	25.5	$3.6{\times 10}^{4}$	${3.1\times 10}^{3}$	${6.6\times 10}^{-2}$
		Decay	17.8	$3.6{\times 10}^{4}$	${4.0\times 10}^{2}$	${1.8\times 10}^{-2}$

^1^ C: Concentration (ppm) and ^2^ RMSE: Root Mean Square Error (kΩ).

**Table 9 sensors-24-04560-t009:** The parameters for the rise and decay stages for the SnO_2_-Fe sensor using the model at different temperatures.

1 ppm	C ^1^	Phenomena	RMSE ^2^	A _(kΩ)_	B	C _(1/t)_
SnO_2_-Fe	80	Rise	23.3	$9.7{\times 10}^{3}$	$1.2{\times 10}^{3}$	$6.6{\times 10}^{-2}$
		Decay	27.3	$9.7{\times 10}^{3}$	$9.9{\times 10}^{1}$	$1.8{\times 10}^{-2}$
	100	Rise	29.2	$7.2{\times 10}^{3}$	$4.1{\times 10}^{3}$	$6.6{\times 10}^{-2}$
		Decay	22.7	$7.2{\times 10}^{3}$	$8.6\times 10^{1}$	$1.8{\times 10}^{-2}$
	120	Rise	25.3	$5.2{\times 10}^{3}$	$4.1\times 10^{3}$	$6.6{\times 10}^{-2}$
		Decay	22.6	$5.2{\times 10}^{3}$	$9.0{\times 10}^{1}$	$1.8{\times 10}^{-2}$

^1^ T: Temperature (°C) and ^2^ RMSE: Root Mean Square Error (kΩ).

## Data Availability

The data will be made available upon request.

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
