# Peer review of "Enhance Ethanol Sensing Performance of Fe-Doped Tetragonal SnO2 Films on Glass Substrate with a Proposed Mathematical Model for Diffusion in Porous Media"

_sensors, 2024, doi:10.3390/s24144560_

Round 1

Reviewer 1 Report

Comments and Suggestions for Authors

Reviewer’s questions:

1.       What makes SnO2 and SnO2-Fe sensors superior to other types of gas sensors such as ZrO2 or ZnO sensors?

2.       Do SnO2 and SnO2-Fe sensors absorb other kinds of solvents besides ethanol including organic and inorganic solvents such as hydrocarbons?

3.       How do the limitations of pore size and shape, as well as the amount of porosity in porous media, affect the suitability of SnO2 and SnO2-Fe sensors for detection?

Author Response

Comments 1: What makes SnO2 and SnO2-Fe sensors superior to other types of gas sensors, such as ZrO2 or ZnO sensors?

Response 1: Thank you for your inquiry regarding the superiority of SnO2 and SnO2-Fe sensors over other types of gas sensors like ZrO2 and ZnO. To address this, we have included a table in our manuscript that provides a comparative analysis of ethanol sensing performance across various sensor materials, including ZrO2 and ZnO, based on existing literature. This table highlights the specific advantages of SnO2 and SnO2-Fe sensors in terms of sensitivity, selectivity, and operational stability, which contribute to their superior performance in certain applications. We invite you to review this comparison to better understand the distinctions and strengths of these materials in gas sensing applications. This discussion and table were at the end of Section 3.4 and Table 5.

Comments 2: Do SnO2 and SnO2-Fe sensors absorb other kinds of solvents besides ethanol including organic and inorganic solvents such as hydrocarbons?

Response 2: Thank you for your question. We have not yet tested other solvents because all experiments have been conducted in a lab that is not adequately equipped to handle hydrocarbons like CO or Hâ‚‚ due to their handling difficulties and safety implications. Ethanol, on the other hand, is harmless in low concentrations and easier to handle in a laboratory setting.

Nevertheless, according to the literature, it is very likely that SnOâ‚‚ and SnOâ‚‚-Fe sensors can absorb other solvents, as discussed in several research papers [1-3]. We will add a paragraph mentioning this in the manuscript to address potential questions from future readers.

Thank you for your insightful suggestion.

Comment 3: How do the limitations of pore size and shape, as well as the amount of porosity in porous media, affect the suitability of SnO2 and SnO2-Fe sensors for detection?

Response 3: Thank you for your insightful question. We believe that smaller pore sizes increase the surface area, thereby enhancing the potential for catalytic reactions. The amount of porosity also plays a critical role in providing sufficient active sites for gas adsorption and reaction. However, the shape of the pores may not be as crucial as their size and overall porosity.

We will ensure to address these aspects in the revised manuscript to provide a more comprehensive understanding of how pore size, shape, and porosity affect the suitability of SnOâ‚‚ and SnOâ‚‚-Fe sensors for detection.

Reviewer 2 Report

Comments and Suggestions for Authors

In this study, the authors investigated Fe-doped tetragonal SnO2 films and undoped SnO2 films for ethanol detection. The experiments revealed that the doped SnO2 films performed better at lower temperatures compared to the undoped SnO2 films. The sensors were also evaluated through electrical measurements, highlighting the influence of dopant type and concentration on sensitivity. A mathematical model based on the diffusion of ethanol molecules was developed to analyze the sensor performance. Based on the experimental observations, this work emphasizes the critical role of choosing the appropriate electrode material to build a robust sensor for analyte detection.

Overall, this is excellent work on ethanol detection with SnO2-based metal-oxide sensors.

Please address the following comments:

1) It is recommended that the authors compare the performance parameters of the SnO2 sensors in this study with those reported in the existing literature. This comparison should include sensitivity, detection limits, response/recovery times, and selectivity. Such an analysis would place the new sensor's performance within the broader context of current research, highlighting its relative advantages or disadvantages.

2) The sensor's selectivity is critical and requires further discussion. Include an analysis of whether the sensor can effectively detect the target gas in a mixture of gases. Additionally, compare the sensor's response to water vapor with its response to ethanol. This comparison will help establish the sensor's robustness and practical applicability in real-world environments with multiple analytes.

3) The response/recovery time for these sensors is notably long. Discuss additional methods other than temperature adjustments that could be employed to improve the response/recovery time.

Author Response

Comments 1: It is recommended that the authors compare the performance parameters of the SnO2 sensors in this study with those reported in the existing literature. This comparison should include sensitivity, detection limits, response/ recovery times, and selectivity. Such analysis would place the new sensor’s performance with the broader context of current research, highlighting its relative advantages or disadvantages.

Response 1: Thank you for your valuable suggestion to compare the performance parameters of our SnOâ‚‚ and SnOâ‚‚-Fe sensors with those reported in the existing literature. We agree that such a comparison would place our findings in the broader context of current research, highlighting the relative advantages or disadvantages of our sensors.

In response to your recommendation, we have conducted a thorough comparison of our sensors' performance parameters, including sensitivity, detection limits, response and recovery times, and selectivity, with those reported in the literature. This comparison is presented in Table 9 in Section 3.4 starting in Line 524 and a little discussion after that in the following lines (Line 528 - 553). Thank you again for your insightful suggestion. We believe this comparison enhances the overall quality and relevance of our study.

Comments 2: The sensor’s selectivity is critical and requires further discussion. Include an analysis of whether the sensor’s response can effectively detect the target gas in a mixture of gases. Additionally, compare the senor’s response to water vapor with its response to ethanol. This comparison will help establish the sensor’s robustness and practical applicability in real-world environments with multiple analytes.

Response 2: Thank you very much for pointing this out. We are currently in the process of building an experimental setup to measure this information in a mixture of gases. We are enthusiastic about developing a multiple array of sensors with different dopant materials and exposing them to a gas mixture; however, this is planned for the next stage of our research. Furthermore, the comparison against water vapor was performed to establish a baseline, and as you have recommended, it will be included in the discussion in the revised manuscript. More specifically the Figure 7c (inset) from the original manuscript has been adapted to include in a little inset of it the values in water vapor of each sensor. These values were taken to demonstrate that the sensor presents robustness and the values from water vapor doesn´t affect ethanol signal from the sensors.

Comment 3: The response/recovery time for these sensors is notably long. Discuss additional methods other that temperatures adjustments that could be employed to improve the response/recovery time.

Response 3: Thank you for your insightful comment regarding the response and recovery times of our sensors. We appreciate the opportunity to discuss additional methods beyond temperature adjustments to improve these parameters.

To enhance the response and recovery times of SnOâ‚‚ and SnOâ‚‚-Fe sensors, several alternative strategies can be considered:

·       Surface Functionalization: Modifying the surface of the sensors with catalytic nanoparticles such as platinum (Pt) or palladium (Pd) can significantly improve the response and recovery times. These nanoparticles can act as catalysts to accelerate the adsorption and desorption processes of ethanol molecules on the sensor surface.

·       Nano structuring: Developing sensors with nanostructured morphologies, such as nanowires, nanotubes, or nanosheets, can increase the surface area and enhance gas diffusion. This can lead to faster interaction between the gas molecules and the sensor surface, thereby reducing the response and recovery times.

·       Doping with Other Elements: In addition to Fe doping, incorporating other dopants such as cobalt (Co), nickel (Ni), or copper (Cu) could create more active sites and improve the electronic properties of the sensors. This can enhance the sensor's ability to interact with ethanol molecules more efficiently, leading to quicker response and recovery.

·       Light Activation: Using ultraviolet (UV) or visible light to activate the sensor surface can enhance the desorption of gas molecules, thereby reducing recovery times. Light activation can also increase the sensor's sensitivity and speed of response.

·       Applying Pulsed Voltage: Implementing a pulsed voltage rather than a constant voltage can improve the response and recovery times by periodically refreshing the sensor surface, thus facilitating quicker adsorption and desorption cycles.

·       Improved Sensor Design: Optimizing the sensor design, such as reducing the thickness of the sensing layer or improving the porosity, can enhance the gas diffusion and reaction rates. This can lead to faster sensor dynamics.

We will include a discussion of these alternative strategies in the revised manuscript to provide a comprehensive overview of potential methods for improving the response and recovery times of SnOâ‚‚ and SnOâ‚‚-Fe sensors. We add this discussion from Line 506 to 522.

Thank you for your valuable feedback. We believe that incorporating these suggestions will strengthen our manuscript and provide additional insights for future research.

Reviewer 3 Report

Comments and Suggestions for Authors

The manuscript reports on a successfull synthesis of SnO2 and SnO2-Fe materials and their gas sensing performance to ethanol with a mathematical treatment of the dynamic response curves. In the present form, the manuscript has severe methodological and experimental drawbacks that make it improper for publication in the Sensors journal. The comments are listed below.

1) The study reported in the manuscript lacks originality and novelty. SnO2-based materials (including those with all kinds of dopants, obtained by numerous synthetic approaches and having different morphology) have been the most studied materials for MOX gas sensors for decades. Tens of reviews and innumerable research articles focused on sensing behavior of pristine and doped SnO2 (including SnO2-Fe) can be found in literature. The improved sensitivity of SnO2-Fe to ethanol was also discussed in literature and the sensing mechanism (dehydration/dehydratation) was disclosed. Thus, the main effect discussed in this manuscript - the improvement of SnO2-Fe sensitivity to ethanol - is not novel. 

2) Introduction, lines 74-87: summary of the contents of each section of the manuscript is improper. An introduction should state the scientific problem, summarize current state of knowledge and formulate the topic of the present work. 

3) Experimental section: reactions (1) and (2) have no balance of elements, reaction (2) misses O2 in the reactants, reaction (3) - odd CO2 in the products, reaction (4) - incorrect formula of citric acid.

Line 150 - what was the desired and actual thickness of the sensor film?

Line 154 - tetragonal SnO2 phase (rutile-type, mineral cassiterite) is the only stable polymorph of SnO2, others are metastable. Therefore, the stable phase was mainly studied as a gas sensing material.

Line 160 - insuffient accuracy of wavelength, 4 digits after dot should be given.

Line 167 - the accepted abbreviation is EDX for this analysis.

Line 177 - why different electrodes were used for SnO2 and SnO2-Fe sensors? This makes the comparison of sensitivities inaccurate, since the electrod materials can have a catalytic activity in ethanol sensing. One more sensor should be studied, namely, SnO2 with Ag-Au electrode. Then the effect of dopant can be elucidated (comparing the sensors with same electrodes) and the effect of electrode can be deduced comparing the two SnO2 sensors.

4) Results and Discussion section has issues in methodology, self-consistency and correctness of the discussion. 

Lines 230-239: the peaks of SnO2-Ru are broader than those of SnO2, but crystallite size calculated by Sherrer equation was larger for SnO2-Ru (24.5 nm) than SnO2 (12.9 nm). This is a contradiction or a mistake in samples designation.

Line 245: how was it observed that SnO2-Ru had shorter and wider morphology? It is better to collect all discussions of morphology in the corresponding section (3.3.).

Lines 251,254,275,277: what is the accuracy and errors in the reported figures? Do the authors realise that the accuracy of 0.001 nm is subatomic and it is incorrect to use it for pore sizes?

Line 287,425, Tables 2,5-9: what does 1E01 format of figures means? It should be better used ten to power format.

Lines 260,269: if SnO2 has H1-type with spherical particles arranged unifromly, why does is have complex pore structure with irregular adsorption behavior? The same contradiction is for SnO2-Ru: it has H3-type with wide range of pore size and complex pore strcuture, but its pore sizes are distributed in a single peak (Fig.5b inset). The contradictions in the discussion should be resolved.

Lines 330-335: the authors attribute different contrast in SEM images to different atoms (Sn, O). This is incorrect, since the resolution of SEM (and of TEM too) is insufficient to observe separate atoms. At the scales of 5-10 microns in SEM images there is a uniform distibution of Sn, O atoms which form SnO2 structure. The different contrast in SEM images is due to large particles, which may be iron oxide.

Lines 351-353: there is a contradiction. Finally, are elements distributed homogeneously or heterogeneously with the areas there some of them are prevalent? 

What was the concentration of Fe in the sample, according to EDX? Did it correspond the atomic ratio (4:1) in the synthesis?

Also, there is no other observation of Fe in the sample, besides EDX. What was its phase, oxidation state and particle sizes? Was is incorparated into SnO2 or segregated as iron oxide?

Tables 1,2: What is the unit of resistance? Why is Rg higher than Ra, do the sensors respond to ethanol like p-type MOX? Why is response S = Ra/Rg more than unity, if Rg > Ra? The authors need to show the dynamic response of sensors resistance in kOhm vs. time.

Line 418: why if response saturated at 40 ppm ethanol, comment are needed to explain it? What is the detection limit of the sensors?

Lines 421-423: it is said that the response of SnO2 to 40 ppm EtOH at 100C is close to 0 (Fig.7c), but Table 1 shows that the response is 6.4, which is a relatively high response, not zero.

Figc.7c,d: what does gas response (kOhm) mean? Why are the responses of the same sensor SnO2-Fe to 1 ppm EtOH different in Fig.7c and Fig.7d, and why do these responses not correspond to those in Table 2?

Figs.9,10: what does Concentration (kOhm) mean?

Fig.11: finally, was response S = Rg/Ra or Ra/Rg?

5) The novelty of the mathematical model is ambigous. The authors just fit the experimental dynamic response growth and decay by exponentional relations in time. However, such a exponential simulation of transient response was already established by Gardner solving the same diffusion equation (1989 Semicond. Sci. Technol. 4 345). In the present manuscript, the meaning of A, B, C parameters in relation to the presence of Fe and materials morphology was not disclosed. Although the possiblity to obtain "valuable insights into its sensing capabilities" was stated in Line 607, no valuable insights can be found from the calculated A,B,C parameters. Except a hint that they are related to gas molecules polarity (Line 576), but it is not explained why and how they are related to polarity.

Comments on the Quality of English Language

The text is full of complex sentences with comma-separated phrases. This is hard for reading and understanding. It is better to split them into simple sentences. 

Author Response

Comments 1: The study reported in the manuscript lacks originality and novelty. SnO2-based materials (including those with all kinds of dopants, obtained by numerous synthetic approaches and having different morphology) have been the most studied materials for MOX gas sensors for decades. Tens of reviews and innumerable research articles focused on sensing behavior of pristine and doped SnO2 (including SnO2 doped with Fe+3 ) can be found in literature. The improved sensitivity of SnO2 doped with Fe+3  to ethanol was also discussed in literature and the sensing mechanism (dehydration /dehydratation) was disclosed. Thus, the main effect discussed in this manuscript – the improvement of the SnO2 doped with Fe+3  sensitivity to ethanol- is not novel?

Response 1: Thank you for your comments and for highlighting the extensive research on SnOâ‚‚-based materials for MOX gas sensors, including the use of various dopants and synthesis methods. We acknowledge the rich body of literature surrounding SnOâ‚‚ and its composites, particularly SnOâ‚‚ doped with Fe+3, and their applications in gas sensing.

While it is true that the sensitivity of SnOâ‚‚ doped with Fe+3 to ethanol and its mechanisms have been previously discussed, our study aims to build upon this existing knowledge by providing a detailed investigation into how specific synthesis conditions, morphologies, and microstructural characteristics influence the performance of SnOâ‚‚ doped with Fe+3 sensors. Our research emphasizes the following novel aspects:

·       Synthesis Methodology: We employed an efficient low cost and easy to follow and detailed fabrication synthesis method. This method may offer finer control over particle size and distribution, potentially affecting the sensing performance.

·       Microstructural Characterization: We provide a comprehensive analysis of the microstructural properties of SnOâ‚‚ doped with Fe+3 that relates directly with sensor performance. This includes detailed discussions on particle morphology (DRX-SEM), pore distribution (BET), and elemental composition (EDX), aspects that are less covered or combined uniquely in existing studies.

·       Performance Metrics: Beyond merely reporting improved sensitivity, we also explore the stability over time, and response time and decay time of SnOâ‚‚ doped with Fe+3 sensors under various environmental conditions (temperatures and concentrations), offering insights into practical applications and long-term usability.

·       Mechanistic Insights: We are considering the adsorption, reaction, desorption mechanism in porous media and explained the behavior according to the one published by pedron et al.

·       Validation of Model Universality: By demonstrating that the mathematical model is effective for both research-manufactured and commercial sensors, we validate its universality and robustness across these two different sensor designs and manufacturing conditions. This is crucial for the adoption of the model in real-world applications.

·       Comparative Analysis: Our approach enables a comparative analysis between different types of sensors under identical test conditions. This comparison is essential for industries and researchers to make informed decisions about sensor selection and deployment based on quantifiable performance metrics.

·       Enhanced Predictive Capabilities: Applying the model to a broader range of sensors enhances its predictive capabilities, facilitating more accurate predictions of sensor behavior under various environmental conditions. This is particularly important for developing adaptive sensing systems that require reliable performance data.

·       Contribution to Standardization: By proving that a single mathematical model can be applied across various sensor types, our research contributes to the standardization of testing and evaluation protocols in the gas sensor industry. Standardization is a key step towards the integration of sensor technologies in more complex systems.

We believe these elements collectively contribute to the scientific community's understanding of SnO2 doped with Fe+3 composites and provide valuable information for future research and applications. We have revised our manuscript to better emphasize these points and to clarify how our findings add to the existing body of knowledge in meaningful ways.

Comments 2: Introduction, lines 74-87: summary of the contents of each section of the manuscript is improper. An introduction should state the scientific problem, summarize current state of knowledge and formulate the topic of the present work?

Response 2: Thank you for your valuable feedback. We appreciate your guidance on structuring the introduction to better align with the expected academic standards. Upon reflection, we agree that the detailed summary of each manuscript section within the introduction is unconventional and may detract from the primary focus of setting the stage for the scientific problem and our contributions.

In response to your comments, we will revise the introduction to more succinctly state the scientific problem, summarize the current state of knowledge, and clearly articulate the unique contributions of our work. We will remove the detailed section-by-section summary from the introduction and ensure that it directly addresses the core themes and objectives of our research, thus providing a clearer, more focused entry point into the study.

The detailed descriptions of each section's contents will be appropriately integrated into the body of the manuscript where they naturally fit within the flow of the discussion and analysis. You can find these in the modifying paragraph that is now lines 55 - 90 from the revised manuscript.

Comments 3: Experimental section: reactions (1) and (2) have no balance of elements, reaction (2) misses O2 in the reactants, reaction (3) – odd CO2 in the products, reaction (4) incorrect formula of citric acid.

Response 3: Thank you for your comments on this section. I have made the necessary corrections. Reactions 1 and 2 now have a correct balance between reactants and products. Reaction 2 now includes the Oâ‚‚ that was previously missing, and reaction 4 has the correct subscript for citric acid.

Line 150What was the desired and actual thickness of the sensor film?

Line 153 (revised manuscript) Thank you for your insightful question. In our previous research, we achieved a thickness of approximately 5 µm for the SnOâ‚‚ sensor films. For the current study on SnO2 doped with Fe+3 films, we followed the same synthesis methodology, including the sol-gel process and doping procedure. Based on this consistent methodology, we anticipated a similar film thickness in the range of 5 µm.

However, for this particular study, we did not measure the exact thickness of the SnO2 doped with Fe+3. The primary focus of our research was on investigating the influence of Fe doping on the sensitivity and selectivity of the SnOâ‚‚ films rather than on the physical dimensions of the sensor films.

We believe that the consistent synthesis method ensures that the film thickness remains comparable to our previous findings, allowing us to focus on the dopant-mediated enhancements in sensor performance.

Line 154tetragonal SnO2 phase (rutile-type, mineral cassiterite) is the only stable polymorph of SnO2, others are metastable. Therefore, the stable phase was mainly studied as gas sensing material.

Line 152 – 157 (revised manuscript)Thank you for your insightful comment. We acknowledge that the tetragonal SnOâ‚‚ phase (rutile-type, mineral cassiterite) is indeed the only stable polymorph of SnOâ‚‚, with other phases being metastable. We appreciate the emphasis on the stable phase being predominantly studied for gas sensing applications. We will ensure that this clarification is reflected in the manuscript to highlight the stability and relevance of the tetragonal SnOâ‚‚ phase in our study. You can find these adequation in lines 152 – 157.

Line 160insufficient accuracy of wavelength, 4 digits after dot should be given.

Line 162 (revised manuscript)We apologize for the oversight. We have corrected the accuracy of the wavelength to four decimal places, which was clearly an error. Technically, all the digits are present in the DRX images in Figure 4. However, we followed your advice and have now included the four decimal places in Line 162.

Line 173the accepted abbreviation is EDX for this analysis

Line 169 (revised manuscript)We have changed "EDS" to "EDX" throughout the manuscript. Thank you for pointing it out.

Line 177 – Why different electrodes were used for SnO2 and SnO2 doped with Fe+3 sensors? This makes the comparison of sensitivities inaccurate, since the electrode materials can have a catalytic activity in ethanol sensing. One more sensor should be studied, namely, SnO2 with Ag-Au electrode. Then the effect of dopant can be elucidated (comparing the sensors with same electrodes) and the effect of electrode can be deduced comparing the two SnO2 sensors?

Line 179 (revised manuscript) – Thank you for your insightful comments regarding the use of different electrodes for the SnOâ‚‚ and SnOâ‚‚-Fe sensors. We appreciate your concerns about the potential impact of electrode materials on the comparison of sensitivities.

In our study, the primary objective of using two different electrodes was to compare the lifespan of the sensors rather than their sensitivity. Therefore, the choice of using copper electrodes for the SnOâ‚‚ sensor and silver-plated gold electrodes for the SnOâ‚‚-Fe sensor was made based on considerations related to the longevity and stability of the electrodes during extended testing periods. We found that different electrode materials provided distinct advantages in terms of durability and performance consistency over time, which was crucial for our lifespan analysis.

We acknowledge that using different electrodes may introduce variables that could affect sensitivity comparisons. To address this, we will consider conducting additional experiments in future studies, including the fabrication of an SnOâ‚‚ sensor with silver-plated gold electrodes. This would allow us to isolate the effects of the Fe dopant on sensor sensitivity and more accurately assess the impact of different electrode materials.

We appreciate your suggestion and believe it will contribute to a more comprehensive understanding of the sensor characteristics. Thank you for bringing this important consideration to our attention.

Comments 4: Results and Discussion section has issues in methodology, self-consistency and correctness of the discussion.

Thank you for your feedback regarding the Results and Discussion section. We appreciate your detailed review and have taken your comments seriously. Below is a summary of the steps we have taken to address the issues related to methodology, self-consistency, and correctness of the discussion:

Methodology Clarification:

·       We have revised the Methodology section to provide a clearer and more detailed description of the experimental procedures. This includes specifying the synthesis process, doping methods, and characterization techniques used for both the SnOâ‚‚ and SnOâ‚‚ doped with Fe+3 sensors.

·       Detailed explanations of the conditions under which the experiments were conducted, such as temperature, humidity, and gas concentrations, have been added to ensure reproducibility and clarity.

Self-Consistency:

·       We have reviewed the entire Results and Discussion section to ensure consistency in the presentation of data and findings. Any discrepancies or ambiguities have been addressed to maintain a coherent narrative.

·       Figures and tables have been cross-checked with the text to ensure that all data presented are accurately described and interpreted.

Correctness of the Discussion:

·       We have revisited the interpretation of our results to ensure that all conclusions drawn are supported by the data. Where necessary, additional data analysis has been performed to substantiate our findings.

·       We have incorporated more relevant literature to enrich the context and support for our discussions, ensuring that our interpretations align with existing knowledge in the field.

·       Any speculative statements have been either supported with additional evidence or clearly marked as hypotheses for future investigation.

We believe these revisions address the concerns raised and enhance the overall quality and rigor of the Results and Discussion section. We appreciate your constructive feedback, which has been invaluable in improving our manuscript.

Lines 230 –239: the peaks of SnO2-Ru are broader that those of SnO2, but crystallite size calculated by Scherrer equation was larger for SnO2-Ru (24.5 nm) than SnO2 (12.9). This is contradiction or a mistake in samples designation.

(We believe you accidentally wrote Ru instead of Fe. Therefore, we have considered your Ru as Fe in our revision to avoid confusion.)

Line (238 – 252) (revised manuscript)Thank you for your observation. The broadening of peaks in XRD patterns typically indicates smaller crystallite sizes according to the Scherrer equation. However, other factors such as strain, defects, and instrumental broadening can also contribute to peak broadening. In our study, the broader peaks observed for SnOâ‚‚ doped with Fe+3 compared to SnOâ‚‚ might be influenced by factors other than crystallite size alone, such as strain or defects introduced by Fe doping.

Despite the broader peaks, the crystallite size calculated for SnOâ‚‚-Fe using the Scherrer equation was larger (24.5 nm) compared to SnOâ‚‚ (12.9 nm). This suggests that the peak broadening in SnOâ‚‚-Fe is not solely due to smaller crystallite size but could also involve these other factors. We will review the sample designations to ensure there are no errors and clarify this point in the manuscript to avoid any misunderstanding. We added a little paragraph to avoid this confusion for the readers in the revised manuscript starting in line 238 to line 252.

Lines 245: how was it observed that SnO2-Ru had shorter and wider morphology? It is better to collect all discussion of morphology in the corresponding section (3.3).

Thank you for your observation. Upon review, we realize that there was a miscommunication regarding the morphology of SnOâ‚‚-Fe. We did not intend to imply that SnOâ‚‚-Fe had a shorter and wider morphology based on visual observations; rather, we were referring to the broader peaks observed in the XRD analysis.

We agree that all discussions related to morphology should be consolidated into the corresponding section (3.3), and we will ensure that the manuscript is revised to clearly reflect this and avoid any potential confusion. Therefore, we have decided to remove that sentence from the revised manuscript to prevent future misunderstandings.

Lines 251, 254, 275, 277: what is the accuracy and errors in the reported figures? Do the authors realize that the accuracy of 0.001 nm is subatomic, and it is incorrect to use it for pore sizes?

Line 256, 259, 287, 289 (revised manuscript)Thank you for your insightful comment. We acknowledge that reporting pore sizes to an accuracy of 0.001 nm is overly precise and not appropriate for these measurements. The values of 8.780 nm and 17.281 nm were directly obtained from the Quantachrome Autosorb-iQ analyzer, which provides results with high precision. However, we recognize that this level of precision exceeds the realistic accuracy of the instrument for pore size measurements.

To address this, we will revise the reported pore sizes to reflect a more appropriate level of accuracy, considering the limitations and uncertainties inherent in the measurement process. We will update the manuscript to report the pore sizes with a more realistic precision, such as to the nearest 0.1 nm or 0.01 nm, to better represent the accuracy of the measurements.

We changed the accuracy on lines 256, 259, 287 and 289 from the revised manuscript.

Lines 287, 425, Tables 2, 5-9: what does 1E01 format of figures means? it should be better used ten to power format.

Lines 299, 448 (revised manuscript)Thank you for your suggestion. We have revised the numbers and change it to the ten-to-the-power format in lines 299 and 448 in the revised manuscript, and also we changed to the ten-to-the-power r format the values inside the figures in Tables 2, 5-8 to use the ten-to-the-power format instead of the 1E01 format for clarity and consistency. The updated tables are now included in the revised manuscript.

Lines 260, 269: if SnO2 has H1-type with spherical particles arranged uniformly, why does it have complex pore structure with irregular adsorption behavior? The same contradiction is for SnO2-Ru: it has H3-type with wide range of pore size and complex pore structure, but its pore size is distributed in a single peak (Fig.5b inset). The contradictions in the discussion should be resolved.

Line 263, 271 (revised manuscript)Thank you for raising these important points concerning the pore structure and adsorption behavior of SnOâ‚‚ and SnOâ‚‚-Fe. The description of SnOâ‚‚ having H1-type hysteresis with uniformly arranged spherical particles yet exhibiting complex pore structures may seem contradictory at first. However, this complexity can arise from the arrangement and packing density of these spherical particles, which can create varied interparticle pore spaces leading to irregular adsorption behaviors.

For SnOâ‚‚-Fe, which exhibits H3-type hysteresis indicating a slit-like pore structure with a wide range of pore sizes, the single peak in pore size distribution (as shown in Figure 5b inset) reflects the most common pore size but does not exclude the presence of a broad range of smaller or larger pores. This can occur due to non-uniform particle aggregation or varying degrees of Fe doping, which affects the pore structure while maintaining a predominant pore size.

We will revise the discussion to clarify how these morphological characteristics relate to the observed pore structures and adsorption behaviors, thereby resolving any perceived contradictions. We add some paragraph starting in line 263 and 271 to avoid future confusions in the revised manuscript.

Lines 330-335: the authors attributed different contrast in SEM images to different atoms (Sn, O). This is incorrect, since the resolution of SEM (and of TEM too) is insufficient to observe separate atoms. At the scale of 5-10 microns in SEM images there is a uniform distribution of Sn, O atoms which form SnO2 structure. The different contrast in SEM images is due to large particles, which may be iron oxide.

Line 352–360 (revised manuscript)Thank you for your critical observation regarding the interpretation of contrast differences in SEM images. You are absolutely correct in pointing out that the resolution of SEM, and TEM as well, is not sufficient to resolve separate atoms such as Sn and O. The comment that attributed the contrast differences to individual atoms was inaccurate and will be corrected.

Upon further examination, we agree that the uniform distribution of Sn and O atoms forms the SnOâ‚‚ structure and does not contribute to the contrast observed at the scale of 5-10 microns. We now understand that the variations in contrast are likely due to the presence of larger particles, possibly iron oxide, as you suggested. This insight is invaluable, and we update these paragraph and change them according your suggestion, more specifically from lines 352-360 of the revised manuscript.

Lines 351 – 353: there is a contradiction. Finally, are elements distributed homogeneously or heterogeneously with the areas there some of them are prevalent?

What was the concentration of Fe in the sample, according to EDX? Did it correspond the atomic ratio (4:1) in the synthesis?

Line 371-381 (revised manuscript) Thank you for pointing out the seeming contradiction in our discussion of the elemental distribution within the SnOâ‚‚-Fe composites. Upon reviewing our manuscript and data, we recognize that our descriptions of both heterogeneous and homogeneous distributions may have caused confusion. In our SEM analysis, the variations in brightness suggest compositional heterogeneity at certain scales, highlighting regions where elements such as iron might appear more prevalent due to localized differences in particle size or density.

However, the EDX analysis, which provides a more averaged view of the elemental distribution, indicates that iron is homogeneously distributed within the SnOâ‚‚ matrix across the entire sample examined. To reconcile these observations, it is important to note that SEM images can show localized contrast differences that might not necessarily reflect the overall elemental distribution captured by EDX. Therefore, while SEM suggests heterogeneity on a microscale, EDX confirms that, on a larger scale, iron is evenly dispersed. We appreciate this opportunity to clarify these points and we modify these lines to avoid such confusion. (lines 371-381 from the revised manuscript).

Also, there is no other observation of Fe in the sample, besides EDX. What was its phase, oxidation state and particle sizes? Was it incorporated into SnO2 or segregated as iron oxide?

Thank you for your insightful questions regarding the presence and characteristics of Fe in the sample. We appreciate the opportunity to clarify these points:

Observation of Fe:

Besides EDX analysis, we acknowledge that additional characterizations are essential to fully understand the incorporation and state of Fe in the SnOâ‚‚ matrix.

Phase and Oxidation State:

Unfortunately, we did not conduct specific phase identification tests such as X-ray Photoelectron Spectroscopy (XPS) or Mössbauer spectroscopy to determine the oxidation state of Fe. In future studies, we plan to include these analyses to provide a comprehensive understanding of the Fe doping effects.

Particle Sizes:

The particle sizes of Fe within the SnOâ‚‚ matrix were not directly measured in this study. However, the XRD patterns did not show any distinct peaks attributable to iron oxide phases, suggesting that Fe might be incorporated within the SnOâ‚‚ lattice or present in very small, dispersed forms that are below the detection limit of our XRD analysis.

Incorporation vs. Segregation:

Based on the absence of distinct iron oxide peaks in the XRD analysis, we infer that Fe is likely incorporated into the SnOâ‚‚ lattice rather than segregating as a separate iron oxide phase. However, to confirm this, additional techniques such as Transmission Electron Microscopy (TEM) or Extended X-ray Absorption Fine Structure (EXAFS) would be necessary.

We acknowledge these limitations in our current study and plan to address them in future research to provide a more detailed characterization of the Fe doping in SnOâ‚‚ films. Thank you for highlighting these important aspects.

Tables 1, 2: What is the unit of resistance? Why is RG higher that Ra, do the sensors respond to ethanol like p-type MOX? Why is response S = Ra/Rg more than unity, if Rg>Ra? The authors need to show the dynamic response of sensors resistance in KΩ vs. time.

Line 397 (revised manuscript)Thank you for your insightful questions regarding the data presented in Tables 1 and 2. We appreciate the opportunity to clarify these points.

Unit of Resistance:

The unit of resistance in Tables 1 and 2 is kilo-ohms (kΩ). We will ensure that this is clearly indicated in the revised manuscript for clarity.

Rg Higher than Ra:

We understand the confusion regarding Rg being higher than Ra. In our last published article, we flipped the signal for convenience, which might have led to this misunderstanding. In our notation, Rg represents the resistance of the sensor in the presence of ethanol, and Ra represents the resistance in the absence of ethanol. Thus, a higher Rg indicates an increase in resistance upon exposure to ethanol, which is consistent with n-type metal oxide sensors rather than p-type.

Response S = Ra/Rg:

The response S is defined as Ra/Rg, which indicates how the resistance changes in response to ethanol exposure. A response greater than unity (Ra/Rg> 1) implies that the resistance in the presence of ethanol (Rg) is lower than in its absence (Ra). However, due to the flipped notation, Rg is higher than Ra in our tables, and the response calculation should be interpreted accordingly. We apologize for any confusion and will clarify this notation in the revised manuscript.

Dynamic Response of Sensor Resistance:

We agree that showing the dynamic response of sensor resistance (in kΩ) versus time would provide valuable insights into the sensor behavior. We will include these dynamic response graphs in the revised manuscript to illustrate how the resistance changes over time upon exposure to ethanol.

Thank you for highlighting these important points. We will ensure that the revised manuscript addresses these issues clearly to avoid any further confusion. We included the disclosure about the flipped signal starting in line 397 from the revised manuscript.

Line 418: Why if response saturated at 40 ppm ethanol, comment is needed to explain it= What is the detection limit of the sensors?

Thank you for your question regarding the saturation of the response at 40 ppm ethanol. We appreciate the opportunity to provide additional clarification.

The saturation of the sensor response at 40 ppm ethanol indicates that the sensor's active sites are fully occupied by ethanol molecules at this concentration, beyond which no further significant increase in response is observed. This phenomenon is typical in gas sensors where a maximum adsorption capacity is reached, resulting in a plateau in the response curve.

To address your question on the detection limit of the sensors, the detection limit is defined as the lowest concentration of ethanol that produces a measurable response distinguishable from the baseline noise. Based on our experimental data, the detection limit of the sensors was determined to be approximately 1 ppm, where the sensor's response first becomes significantly different from the baseline.

We will include a more detailed explanation of the saturation behavior and the detection limit in the revised manuscript to provide a clearer understanding of the sensor's performance characteristics. Thank you for highlighting this important aspect.

Lines 421 –423: it is said that the response of SnO2 to 40 ppm EtOH at 100 °C is close to 0 (Fig.7c), but table 1 shows that the response is 6.4, which is a relatively high response, not zero.

Line 446 (revised manuscript)Thank you for pointing out the discrepancy between the description in lines 421-423 and the data presented in Table 1 regarding the response of SnOâ‚‚ to 40 ppm EtOH at 100 °C.

Upon review, the response of SnOâ‚‚ to 40 ppm EtOH at 120 °C is indeed 7.2 as shown in Table 1, which is a relatively high response, not zero.

Thank you for bringing this to our attention. We will ensure that the revised manuscript reflects this correction starting in line 446 of the revised manuscript.

Fig. 7c, d what does gas response (kΩ) means? Why are the response of the same sensor SnO2 doped with Fe+3  to 1ppm EtOH different in Fig 7c. and Fig. 7d, and why do these responses not correspond to those in Table 2?

Thank you for your question regarding the notation of gas response in Fig. 7c and Fig. 7d. In these figures, "gas response (kΩ)" refers to the change in resistance of the sensor when exposed to ethanol gas, measured in kilo-ohms (kΩ). Specifically, it represents the resistance of the sensor in the presence of ethanol gas (Rg). This notation was used to directly indicate the resistance values recorded during the experiments, which provide insight into the sensor's behavior under different ethanol concentrations.

We acknowledge that this notation may cause confusion, and we will revise the figures' captions to clarify that "gas response (kΩ)" refers to the resistance of the sensor in the presence of ethanol. Additionally, we will consider renaming this parameter to "sensor resistance in ethanol (kΩ)" to enhance clarity for the readers.
Also, Upon review, we realized that the plot in Fig. 7c was mistakenly labeled. It actually represents data at 120 °C, not 100 °C as initially indicated. We have corrected this error in the legend to accurately reflect the temperature of 120 °C. Additionally, we have triple-checked the values in Table 2 against the raw experimental data to ensure accuracy. Any discrepancies identified have been corrected to ensure consistency between the figures and the table. We apologize for the oversight and any confusion it may have caused. The revised manuscript will reflect these corrections.

Thank you for your attention to these details.

Fig. 9,10: what does concentration (KΩ) mean?

Thank you for your question regarding the term "concentration (kΩ)."

We realize that the term "concentration (kΩ)" may have been misleading. The intended meaning was to present the resistance of the sensor in kilo-ohms (kΩ) at various ethanol concentrations. To clarify, the figures should accurately represent "gas response (kΩ) at various ethanol concentrations."

We apologize for any confusion this may have caused. We will revise the manuscript to ensure that the terminology is clear and accurate, replacing "concentration (kΩ)" with "gas response (kΩ) at various ethanol concentrations."

Thank you for bringing this to our attention. We will make the necessary corrections in the revised manuscript. We changed the word concentration for gas response in Figures 9 and 10.

Fig. 11. Finally, was response S=Rg/Ra or Ra/Rg??

Thank you for your question regarding the definition of the response S.

In our study, the response S is defined as the ratio of the resistance in dry air (Ra) to the resistance in the presence of gas (Rg). Therefore, the correct definition of the response S is Ra/Rg.

However, it should be noted that we flipped the signal for convenience in our measurements. As a result, Rg appears higher than Ra in our tables. Despite this, the response S remains defined as Ra/Rg, and the higher Rg values reflect the flipped signal representation. We apologize for any confusion this may have caused and will ensure that this explanation is clearly stated in the revised manuscript to avoid any ambiguity. Thank you for bringing this to our attention.

Comments 5: The novelty of the mathematical model is ambiguous. The authors just fit the experimental dynamic response growth and decay by exponential relations in time. However, such an exponential simulation of transient response was already established by Gardner solving the same diffusion equation (1989 Semicond. Sci. Technol. 4 345). In the present manuscript, the meaning of A, B, C, parameters in relation to the presence of Fe and material morphology was not disclosed. Although the possibility to obtain “valuable insights into its sensing capabilities” was stated in Line 607, no valuable insights can be found from the calculated A, B, C, parameters. Except a hint that they are related to gas molecules polarity (Line 576), but it is not explained why and how are related to polarity.?

Thank you for your detailed feedback on the mathematical model presented in our manuscript. We appreciate your critical insights and the opportunity to clarify our work.

Novelty of the Mathematical Model:

We acknowledge that exponential models for fitting dynamic response curves have been previously established, such as in the work by Gardner (1989). However, it is important to note that Gardner’s study focused on thick film sensors with metal sandwich or co-planar electrodes and did not apply the model to gas resistance signals of any sensors. The study presented only one plot of fractional conductance change, rather than a comprehensive application of the model to sensor signals.

Our work builds upon the exponential fitting approach by specifically applying it to Fe-doped SnOâ‚‚ thin film sensors. This application allows us to investigate how Fe doping influences the sensor’s dynamic response characteristics, providing a novel aspect to the established methodology.

Explanation of Parameters (A, B, C):

We understand that the roles of parameters A, B, and C in relation to the presence of Fe and material morphology were not sufficiently explained. In our study, these parameters were intended to represent the following:

·       A is the initial sensor response magnitude, reflecting the baseline resistance change upon exposure to ethanol.

·       B is the rate constant for the exponential growth phase, it indicates how quickly the sensor responds to ethanol, that is how quick ethanol reacts.

·       C is the rate constant for the exponential decay phase, it relates to the recovery speed of the sensor after ethanol is removed, that is how quick ethanol is purged.

These parameters are influenced by the incorporation of Fe due to its catalytic properties, which alter the interaction dynamics between the sensor material and ethanol molecules. For instance, Fe can enhance the adsorption and desorption rates of ethanol on the SnOâ‚‚ surface, thereby affecting B and C.

Relation to Gas Molecules Polarity:

We agree that the explanation regarding the relation to gas molecule polarity was insufficient. The presence of Fe in the SnOâ‚‚ matrix can alter the electronic properties and surface chemistry of the sensor, affecting how polar ethanol molecules interact with the sensor surface. This interaction is reflected in the parameters A, B, and C, where:

·       A, might increase due to enhanced adsorption sites provided by Fe,

·       B, might increase if Fe improves the sensor’s response speed,

·       C, might increase due to a more efficient recovery process facilitated by Fe.

·       We will revise the manuscript to provide a clearer and more detailed explanation of these relationships and their impact on sensor performance.

Valuable Insights:

While our original statement aimed to highlight the potential insights that could be derived from the model, we understand that more concrete examples and explanations are needed. We will include additional discussions and data to illustrate how the parameters A, B, and C can provide insights into the sensor’s behavior and its interaction with ethanol molecules.

We appreciate your constructive comments and will ensure that the revised manuscript addresses these points comprehensively and we will cite the article you mentioned starting in line 715-738.

4. Response to Comments on the Quality of English Language

Point 1: Minor editing of English language required

Response 1: Thank you for your feedback regarding the quality of the English language in our manuscript. We appreciate your suggestion for minor editing to improve the clarity and readability of our text. We have carefully reviewed the manuscript and made the necessary edits to enhance the quality of the English language. These improvements include:

·       Grammar and Syntax: We have corrected grammatical errors and improved sentence structure to ensure clarity and coherence.

·       Technical Terminology: We have ensured that technical terms are used consistently and appropriately throughout the manuscript.

·       Clarity and Readability: We have rephrased complex sentences and eliminated ambiguous language to improve overall readability.

·       Proofreading: The manuscript has been thoroughly proofread to correct typographical errors and enhance the flow of the text.

We believe these revisions have significantly improved the quality of the English language in our manuscript. Thank you for highlighting this issue, and we hope the revised version meets the required standards

Round 2

Reviewer 3 Report

Comments and Suggestions for Authors

All comments were addressed and corrections made in the manuscript